# Pan-genome analysis of 13 *Malus* accessions reveals structural and sequence variations associated with fruit traits

Ting Wang[1,4], Shiyao Duan [2,4], Chen Xu[1], Yi Wang[1], Xinzhong Zhang[1], Xuefeng Xu[1], Liyang Chen[3], Zhenhai Han[1] ✉ & Ting Wu[1] ✉

Structural variations (SVs) and copy number variations (CNVs) contribute to trait variations in fleshy-fruited species. Here, we assemble 10 genomes of genetically diverse *Malus* accessions, including the ever-green cultivar 'Granny Smith' and the widely cultivated cultivar 'Red Fuji'. Combining with three previously reported genomes, we assemble the pan-genome of *Malus* species and identify 20,220 CNVs and 317,393 SVs. We also observe CNVs that are positively correlated with expression levels of the genes they are associated with. Furthermore, we show that the noncoding RNA generated from a 209 bp insertion in the intron of mitogen-activated protein kinase homology encoding gene, *MMK2*, regulates the gene expression and affects fruit coloration. Moreover, we identify overlapping SVs associated with fruit quality and biotic resistance. This pan-genome uncovers possible contributions of CNVs to gene expression and highlights the role of SVs in apple domestication and economically important traits.

Agronomically important traits in crop plants are shaped by genetic variation derived from their wild ancestors, as well as the selection and maintenance of mutations that reflect agricultural adaptations and human preferences[1,2]. Broad genetic diversity is of great potential value for breeding new cultivars[3–5]; however, significant evolutionary bottlenecks have arisen in most major crops due to their complex genomes, characterized by large size, high heterozygosity level and polyploidy[6–8]. Most of this variation is quantitative, and a key challenge in plant breeding is to determine how specific genes and variants contribute to quantitative trait variation[9–11].

Structural variations (SVs, including large deletions, insertions, duplications and chromosomal rearrangements) are a major source of genetic variation[12,13]. There is increasing evidence from human studies that SVs are responsible for many human diseases[12], and decades of research have also shown that they are important for plant evolution and agriculture, affecting traits such as shoot architecture, flowering time, fruit quality, and stress resistance[13–15]. However, identifying SVs with short-read sequencing is extremely challenging, typically giving poor resolution and thus obscuring their molecular and phenotypic effects[16,17]. In contrast, high-throughput long-read sequencing now enables a broad survey of population-scale SV landscapes, and this approach allows higher sensitivity and accuracy for detecting SVs[16]. Accordingly, a growing number of studies, based on high-quality genome assemblies, are resolving SVs in diverse plant species[18–30].

Apple is one of the most widely produced and economically important fruit crops in temperate regions. Cultivated apple (*Malus domestica* Borkh.) was domesticated from its wild relatives *M. sieversii* and *M. sylvestris* in the Tian Shan Mountains 4000–10,000 years ago[4]. The first reference genome of apple was assembled for the "Golden Delicious" cultivar, which paves the way for apple functional genomics studies[31]. This was followed by the reports of reference-quality genome assemblies from the double-haploid GDDH13 line[32], the triple-haploid HFTH1 line[33], a diploid cultivar 'Gala Galaxy'[34], and *M. baccata*[9]. However, the availability of only a few reference genomes, that represent only a fraction of the full range of genetic diversity within a species, limits the identification of agronomic traits related to larger SVs[35–41].

[1]College of Horticulture, China Agricultural University, Beijing, China. [2]Plant Science and Technology College, Beijing University of Agriculture, Beijing, China. [3]Smartgenomics Technology Institute, Tianjin, China. [4]These authors contributed equally: Ting Wang, Shiyao Duan. ✉e-mail: rschan@cau.edu.cn; wuting@cau.edu.cn

Recently, an apple pan-genome was constructed with genomes of cultivated apple (*M. domestica* cv. Gala) and its two primary wild progenitors, *M. sieversii* and *M. sylvestris*[24]. It uncovered thousands of genes and identified the introgression of genes/alleles through hybridization during apple domestication. The study also highlighted the potential value of a pan-genome constructed from diverse apple populations to enable fast and accurate genotyping, particularly for larger SVs[24]. Self-incompatible apple (*Malus*) species have particularly high levels of genetic variation, suggesting that heterozygous genomic regions may play an important role for phenotypic variation, which is in contrast to that in self-compatible species[24,42]. Currently, the representative reference genomes of apple lack identified SVs, which prevents an understanding of how they influence key traits, information that would be valuable for apple molecular breeding.

Here, we assemble genomes of 10 apple accessions that exhibit broad diversity in fruit quality and disease resistance. Coupling these with three existing genomes (*M. domestica* cv. Gala, *M. sieversii*, and *M. sylvestris*), we assemble the pan-genome. Analyzes of the pan-genome reveal substantial genetic variations, including SVs and gene fusion events. We also discover that candidate SVs and gCNVs (gene copy number variations) could have shaped environmental adaptations and agronomic traits by modulating gene expression. Our work shows the prevalence and importance of SVs in apple genomes and introduces a suite of resources and tools for apple genetic improvement community.

## Results

### Genome assembly

The 13 apple accessions used for pan-genome assembly include four wild apple species and nine apple cultivars (*M. domestica*). The four wild species are *M. sylvestris* (MSM), *M. sieversii* (MSR), *M. orientalis* (MO), and *M. asiatica* "Zisai Pearl" (MA). The nine cultivator include "Cox's Orange Pippin" (COP), "Red Fuji" (RF), "Gala" (GA), "Granny Smith" (GS), "Honeycrisp" (HC), "Jiguan" (JG), "Orin" (OR), "Ralls" (RA), and "Starking Delicious" (SD). The 13 *Malus* accessions diversified in fruit color, fruit size, stress resistance (Fig. 1a). Of these accessions, genome assemblies of MSM, MSR and GA have been reported previously (PRJNA591623), with haploid consensus sizes of 652–668 Mb, contig N50s of 2.32–18.88 Mb and heterozygosity ratios ranging from 0.85% to 1.28%[24]. Despite the high heterozygosity rates, the three assemblies exhibited a high degree of continuity.

The other ten accessions were sequenced by PacBio HiFi with 27.82–53.15× coverage (Supplementary Table 1). The genome assembly sizes were estimated to be between 668.25–679.75 Mb, and to have heterozygosity rates of 0.88–1.67% and repeat rates of 56.08–59.14% (Supplementary Table 2). The final assembly sizes of these ten genomes were between 661.83–668.75 Mb and the contig N50s[33] between 24.06–39.17 Mb (Table 1; Supplementary Table 2). To generate high-quality assemblies, the GS and FJ genomes were de novo assembled using a combination of PacBio CCS sequencing and chromosome conformation capture (Hi-C) sequencing (Supplementary Fig. 1). We further evaluated the completeness of the assemblies using Benchmarking Universal Single-Copy Orthologs (BUSCO)[43], and found scores between 96.9% (GS) and 98.0% (FJ), which are higher than the corresponding values of a previously published *Malus* genome (HFTH1, PRJNA482033), indicating high completeness within genic regions (Supplementary Tables 3–5). After long terminal repeat (LTR) assembly index (LAI) assessment[44], we found that only the GS genome assembly reached the "gold standard" level (LAI > 20), and therefore qualified as a reference genome (Table 1; Supplementary Table 6). Next, the other eight genome assemblies were aligned to the GS genome, by fragmentation and having short reads mapped against the GS

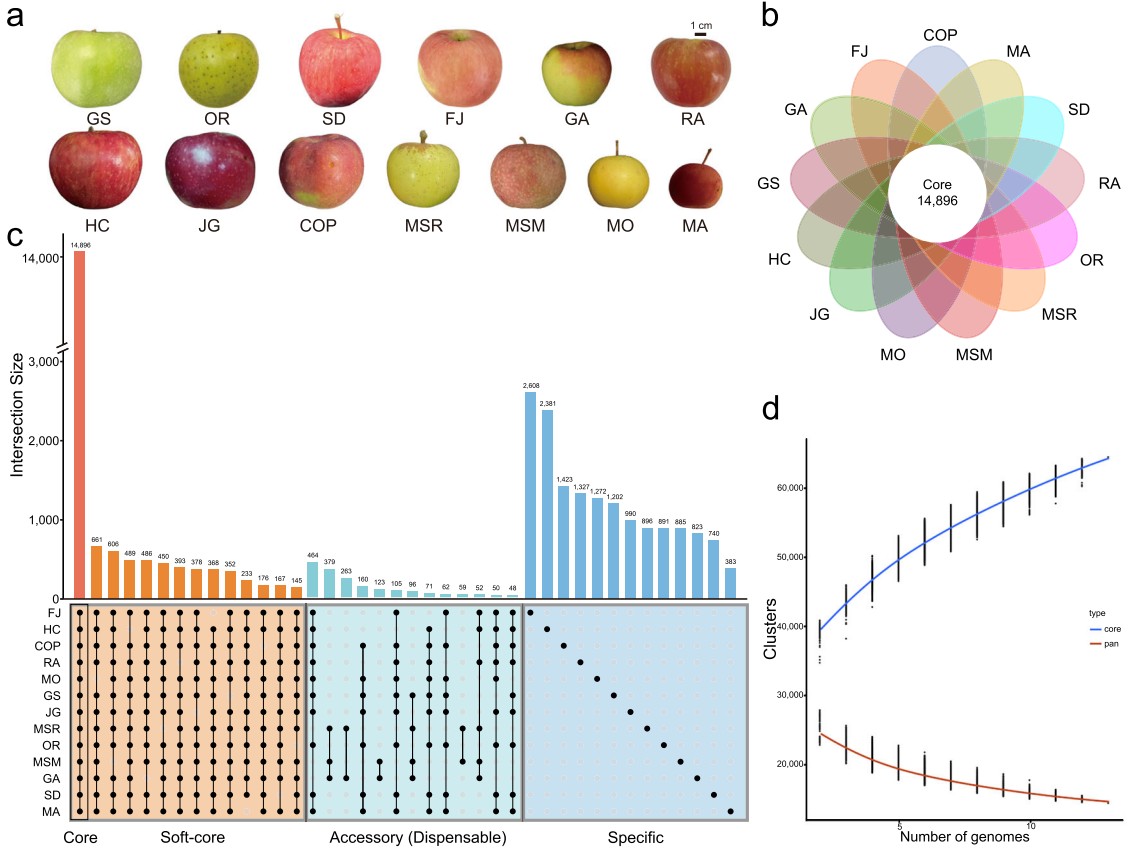

**Fig. 1 | Pan- and core genome analysis of 13 apple accessions. a** Fruit phenotypes of 13 diverse *Malus* accessions. Scale bar 1 cm. **b** Flower plot showing the number of core gene clusters. **c** Gene cluster analysis. **d** Variations in the number of gene clusters between the pan-genome and the core genome.

**Table 1 | Summary statistics from assembly and annotation of ten *Malus* genomes**

| Accession | GS | MO | HC | JG | COP | RA | OR | SD | MA | FJ |
|---|---|---|---|---|---|---|---|---|---|---|
| Assembly | | | | | | | | | | |
| Total length of scaffolds length (Mb) | 664.33 | 665.07 | 661.83 | 663.59 | 668.03 | 662.88 | 668.75 | 662.07 | 662.38 | 665.44 |
| Total number of scaffolds | 960 | 212 | 168 | 53 | 89 | 137 | 54 | 60 | 80 | 361 |
| Scaffold N50 (Mb) | 37.47 | 38.97 | 37.85 | 37.23 | 39.17 | 36.10 | 37.70 | 36.99 | 37.53 | 36.97 |
| Total length of contigs (Mb) | 664.32 | 665.04 | 661.83 | 663.59 | 668.03 | 662.88 | 668.74 | 662.07 | 662.38 | 665.44 |
| Total number of contigs | 1,042 | 485 | 203 | 57 | 104 | 160 | 62 | 73 | 91 | 395 |
| Contig N50 (Mb) | 32.80 | 33.27 | 24.06 | 37.23 | 39.17 | 32.79 | 37.01 | 36.69 | 36.17 | 32.99 |
| Sequences anchored to chromosome (%) | 94.96 | 98.89 | 98.38 | 98.01 | 99.41 | 98.57 | 99.68 | 99.13 | 96.72 | 96.97 |
| Gap counts | 82 | 273 | 35 | 4 | 15 | 23 | 8 | 13 | 11 | 34 |
| Complete BUSCOs (%) | 96.90 | 95.40 | 96.80 | 97.10 | 98.10 | 96.80 | 98.50 | 98.30 | 97.90 | 98.00 |
| LTR assembly index (LAI) | 20.01 | 19.47 | 18.44 | 18.02 | 19.18 | 18.88 | 18.10 | 19.18 | 20.60 | 18.13 |
| Annotation | | | | | | | | | | |
| Percentage of repeat sequences (%) | 56.11 | 56.21 | 55.56 | 56.01 | 55.95 | 55.44 | 55.86 | 55.63 | 55.76 | 54.05 |
| Number of genes | 46,050 | 45,167 | 45,541 | 45,653 | 45,547 | 45,373 | 45,320 | 45,027 | 45,359 | 45,948 |

genome, resulting in chromosomes covering 96.72–99.41% of the genome (Table 1).

The ten apple genomes were evaluated by BUSCO (v2), with each accession achieving a score >95%. The scores ranged from 95.4% (MO) to 98.5% (OR) (Supplementary Table 5; Supplementary Fig. 2a). Using LAI to assess the LTR completeness, we obtained scores of 18.02–20.60 in each of the ten genomes (Supplementary Table 6), close to the "gold standard". Our genomes showed an overall high collinearity between GS and the other nine assemblies (Supplementary Fig. 2b), indicating an excellent continuity and integrity between the assembled genomes.

**Genome annotation and gene-based pan-genome construction**

We determined the sum of the sequence length of each assembled transposable element (TE), which ranged from 359,707,305 to 112,373,813,146 bp, accounting for 54.06% to 56.21% of the total assembled sequence length (Supplementary Data 1; Supplementary Fig. 3a). When the TE protein sequences were classified into retrotransposons (Class I) and DNA transposons (Class II), we found that the proportions were 91% and 9%, respectively. In Class I, LTR retrotransposons were the most common type, followed by Gypsy, Copia (Supplementary Fig. 3b). Among them, MO had the highest TE content of the 10 genomes. The transposon content in the genomes was similar (Supplementary Data 1).

Sequence annotation showed that repeat sequences from the ten genomes accounted for 54.05–56.21% of each genome (Table 1), with the number of annotated genes being between 45,027 and 46,050, of which > 95.5% had annotated functions (Supplementary Table 7; Supplementary Data 2). We collected leaf samples from each accession, and fruit peel and fruit flesh samples from the five apple accessions (GS, OR, SD, FJ, and GA), for a total of 22 samples. RNA sequencing was performed on all 22 samples with a mean of 6 Gbp (Supplementary Table 8). We acquired a total of 465,819,938 clean reads with an average of 21,173,633 reads (range 19,164,404 to 22,590,783) for each sample. The Q20 and Q30 sequencing quality scores were 97.6% and 93.3%, respectively, and the GC content for all samples ranged from 45.2% to 48.6% (Supplementary Table 8).

An apple pan-genome set was constructed using the 13 assembled genomes (Fig. 1b). The number of genes in each genome was similar, ranging from 45,027 to 46,050 (Supplementary Table 9). Gene cluster analysis was conducted with OrthoFinder[26,45,46], which revealed a total of 46,409 clusters containing 590,746 genes. Core clusters were defined as those shared by all 13 genomes. There were 14,896 core clusters containing 287,868 genes, corresponding to 32.10% and 48.72% of the total numbers of clusters and genes, respectively. The

fact that ~1/3 of all clusters were core clusters but closer to half of all genes in each genome were present in core clusters indicated the presence of multi-copy genes derived from whole genome duplication[24]. Furthermore, we constructed a pan-genome using nucleotide sequence-based methods to confirm the number of core genes. Using this method, we identified 22,005 gene loci that were retained in all 13 genomes. These still comprised approximately half of each genome and were termed the core loci.

We further defined clusters that were present in 11 or 12 accessions as soft-core clusters. There were 7,786 of these, containing a total of 123,759 genes (16.78% and 20.95% of the total cluster and gene numbers, respectively). The 23,027 clusters (49.62%) present in 2 to 10 accessions contained 159,297 genes (26.97%) and were defined as accessory (dispensable) clusters; the 700 clusters (1.51%) that were present in only one accession each contained 19,822 genes (3.36%) and were defined as specific clusters (Fig. 1c; Supplementary Table 9). As we sequentially added more genomes to the pan-genome, the number of core genes decreased until a total of 10 genomes was included, after which the number was similar, suggesting that our sequencing data is sufficient to construct a pan genome (Fig. 1d).

A Kyoto Encyclopedia of Genes and Genomes (KEGG) pathway analysis[47] showed that core genes were enriched in pathways related to glycolipid metabolism, such as glycerophospholipid metabolism, and various types of N-glycan biosynthesis. In addition, many core genes were annotated as being related to circadian rhythms. Notably, accessory (dispensable) genes were significantly enriched in flavonoid biosynthesis, suggesting that secondary metabolism has diversified in different apple cultivars (Supplementary Fig. 4). To improve our understanding of the genetic differences between cultivated and wild apple species, we categorized genes into those only present in wild-type apples and those only present in cultivated apples. KEGG analysis was then conducted, and the results showed that two secondary metabolism pathways-phenylpropanoid biosynthesis and terpenoid biosynthesis (including diterpenoid biosynthesis, sesquiterpenoid and triterpenoid biosynthesis) -as well as two stress resistance pathways-ascorbate and aldarate metabolism, and cysteine and methionine metabolism-were significantly enriched in wild-type apple[48–52]. These findings highlight a remarkable difference in composition of secondary metabolites between wild and cultivated apples, which explained the varied stress response between wild and cultivated apples (Supplementary Fig. 5). Moreover, a KEGG analysis comparing cultivars with colored or uncolored fruit, revealed enrichment in the former of genes involved in the biosynthesis of various secondary metabolites, the flavonoid class

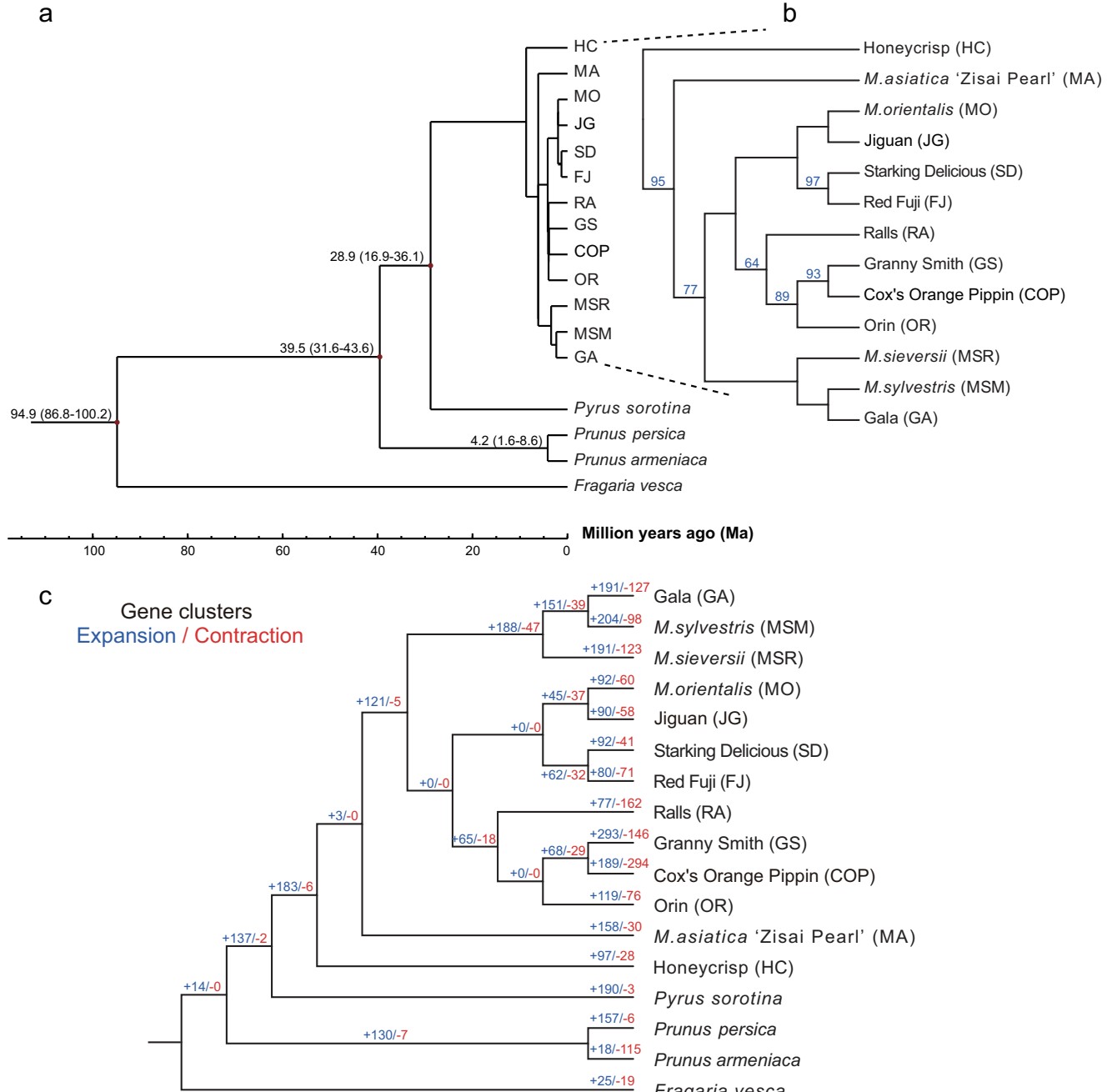

**Fig. 2 | Evolutionary relationships and genetic diversity of apple accessions.**
**a** Phylogenetic tree of 13 *Malus* accessions and 4 outgroup accessions. Black characters are estimated divergence times (Ma) based on single-copy orthologous group inference. **b** Unrooted phylogenetic tree showing the relationships between 13 *Malus* accessions. Numbers represent bootstrap values. **c** The expansion and contraction of gene clusters between 13 *Malus* accessions and four outgroup accessions. Blue characters are the number of expanding gene clusters, red characters are the number of contracting gene clusters.

of phenylpropanoids, which are associated with fruit coloring (Supplementary Fig. 6).

### Evolution of the apple genome

To clarify the phylogenetic relationship and genetic population structure of *Malus* and the other four species, we constructed a phylogenetic tree. *Pyrus* and *Malus* did not diverge until about 28.9 (16.9–36.1) million years ago, during the first epoch (Miocene) of the Neogene (Fig. 2a). As expected, in *Malus*, the four wild-type apples (MA, MSR, MSM and MO) were separated from the accessions of cultivated apples (Fig. 2a; see details of the *Malus* clade in Fig. 2b). MA, which originated in China, clustered into a clearly separated

monophyletic clade from other wild-type apples, with OR and GS with uncolored fruit being closely related (Fig. 2a). FJ is a hybrid progeny of RA × "Delicious", while SD is a bud sport of "Delicious", which is reflected in the phylogeny by their highly supported relationship as sister taxa (Fig. 2a). The divergence time between apple and other species was consistent with previous studies, supporting our findings[53,54].

In addition to species divergence time, we also performed analyses of cluster expansion and contraction, and positive selection to elucidate evolutionary relationships. We identified 53,803 gene clusters in the 13 apple genomes and those of four other species (*Pyrus sorotina, Prunus armeniaca, Prunus persica* and *Fragaria vesca*). There

were significant differences among species, such that 183 and 6 clusters in *Malus*, and 190 and 3 families in *Pyrus* had expanded noticeably and contracted slightly, respectively (Fig. 2c). In apple, GS and COP had high numbers of expanded and contracted clusters: 293 and 146 in GS, and 189 and 294 in COP (Fig. 2c). To improve the understanding of the significance of expanded and contracted gene clusters, we conducted KEGG analysis of expanded and contracted genes in all domesticated cultivars (Supplementary Data 3). Interestingly, we found significant enrichment for the flavone and flavonol biosynthesis, and for the iso-flavonoid biosynthesis pathways in FJ, RA and JG, which produce low levels of antioxidants, and may have an association with the extent of flesh browning[55–57].

Gene evolution was subject to positive selection, often reflecting a remarkable adaptability to the environment. The probability of posi-tive selection was detected by calculating Ka/Ks using the branch model of PAML (v4.8)[58]. The higher Ka/Ks values of genes involved in stress responses shared between MA, MSR, MSM and MO suggested that they had been under stronger evolutionary selection than culti-vated apple (Supplementary Data 4–13).

## Gene CNVs associated with variations in fruit development

CNVs are polymorphisms within species in which sections of a genome differ in copy number between individuals, and include deletions, duplications or amplifications (the same sequence of DNA duplicated multiple times, typically in tandem) of DNA sequence[13]. Our results suggested that repeat sequences and number of genes were similar in apple accessions, which indicated that the divergence between related species is relatively conservative in apple and that apple evolution was not driven by repeat sequences (Table 1). It is clear that CNVs have contributed broadly to plant evolution and domestication; however, CNV analysis remains challenging[13]. We took advantage of our high-quality assemblies and systematically explored agronomic traits rela-ted to CNVs by exploiting alignments of predicted protein sequences of pan-genome genes from the 13 assemblies. In total, 20,220 protein-coding CNV genes were identified, accounting for 43.93% of the apple pan-genome (Fig. 3a; Supplementary Data 14). The gCNV catalog included 4,945 gene deletions (Supplementary Table 10). In addition, 418, 405 and 775 CNV genes were positively correlated with their expression levels in the fruit peel, fruit flesh (for accessions GS, OR, SD,

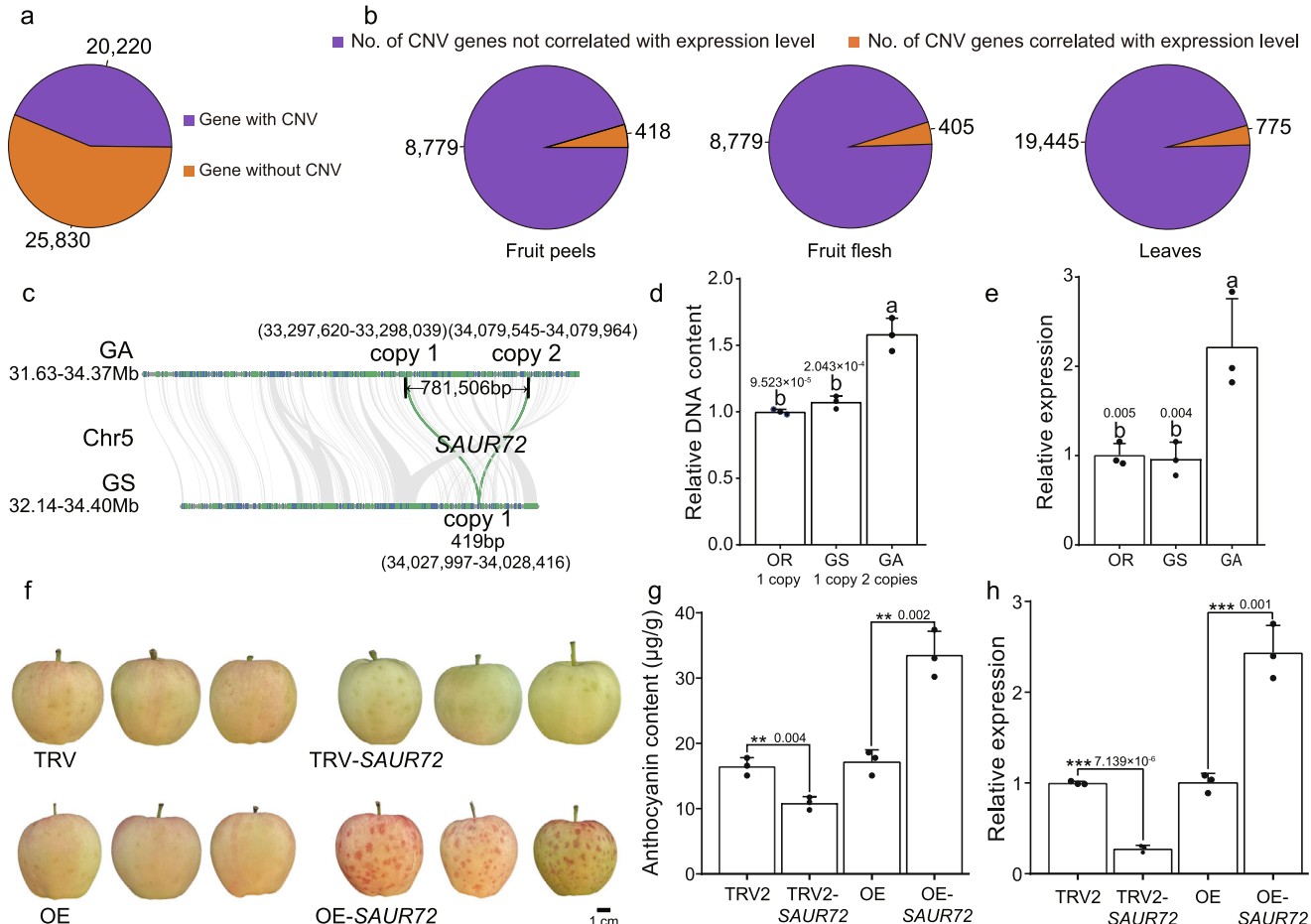

**Fig. 3 | Gene CNVs are widespread and are associated with agronomic trait variation. a** The number of pan-genome genes with or without CNVs across the 13 assemblies. *If two or more genes in the pan-genome shared both identity and coverage > 90%, we only selected one gene for CNV identification. **b** Number and proportion of CNV genes for which copy number was significantly correlated (at $p < 0.05$ and $|r| > 0.5$) with the gene expression level in the fruit peel and fruit flesh (for accessions GS, OR, SD, FJ, and GA) and with the gene expression level in the leaves (for all 13 accessions). **c** Schematic illustration of *SAUR72*. GA and GS represent the apple cultivars *M. domestica* cv. "Gala" and "Granny Smith", respec-tively. **d** DNA qPCR validation of the *SAUR72* copy number. **e** Expression levels of *SAUR72* in different accessions. Data are presented as mean values ± SD of three independent biological replicates. Different letters (a, b) above the bars indicate

significantly different values ($p < 0.05$) calculated using Duncan"s multiple tests. *p* values are shown in (**d** and **e**). **f** Typical phenotypes of *SAUR72* silenced (TRV-*SAUR72*) and overexpressing (OE-*SAUR72*) "Gala" fruit peels. Scale bar 1 cm. **g** Anthocyanin contents of *SAUR72* silenced (TRV-*SAUR72*) and overexpressing (OE-*SAUR72*) "Gala" fruit peels. Data are presented as mean values ± SD of three inde-pendent biological replicates. Asterisks indicate significant difference by Student's *t*-test (two-sided; ** $p < 0.01$). **h** Relative expression of *SAUR72* in *SAUR72* silenced (TRV-*SAUR72*) and overexpressing (OE-*SAUR72*) "Gala" fruit peels. Data are pre-sented as mean values ± SD of three independent biological replicates. Asterisks indicate significant difference by Student's *t*-test (two-sided; ** $p < 0.01$, *** $p < 0.001$), *p* values are shown in **g** and **h**. Source data are provided as a Source Data file.

FJ and GA) and leaves (for all 13 accessions), respectively (Fig. 3b; Supplementary Data 15–17).

We identified a SMALL AUXIN UP RNAs (SAUR) gene, GS05G0207300 (*MdSAUR72*), as a CNV with an expression profile that was closely related to apple fruit development. Auxin is involved in all stages of plant development, and *SAUR* genes constitute the largest family of early auxin response genes. Previous studies have shown that *SAUR72* overexpression leads to early leaf maturity in *Arabidopsis*, suggesting a role for auxin in regulating leaf senescence[59]. In apple fruit peel, we found that *MdSAUR72* had two copies in GA, but only one copy in GS and OR (Fig. 3c). Furthermore, RT-qPCR analysis showed that *MdSAUR72* was more highly accumulated in GA than in GS and OR at both the DNA and RNA level (Fig. 3d, e). When we investigated *MdSAUR72* function by transient overexpression assays in apple peel, we observed increased anthocyanin accumulation and upregulated expression of anthocyanin biosynthesis-related genes, while silencing *MdSAUR72* had the opposite effect (Fig. 3f). These results indicated that duplication of this gene affects apple fruit development, and may be useful in future apple breeding programs.

## Identification of SVs and SV hotspot regions

To improve the reliability and resolution of SV identification, we used three approaches for long-read sequencing: SV detection SyRI (20-06-2022)[60], Assemblytics[61], and SVMU[62]. SVs with an overlap ≥90% using all three approaches were identified as candidate SVs and were used for further analysis[28].

We distinguished between four SV types: inversions, translocations, presence and absence. For each assembly, 72,188–115,748 SVs were identified, with the variation of presence/absence (PAV) being significantly higher than the other types of SVs. PAV was between 64,418-101,533 in the different accessions, indicating that it was the main contributor to genome size variation, while translocations were <14,005 and inversions <210 (Fig. 4a, b; Supplementary Table 11). Results showed that there were few core genes containing PAVs/SVs in the exonic regions, while the non-core genes contained a high proportion of PAVs. Cross-analysis confirmed that PAVs could affect the production of non-core gene clusters, suggesting that non-core genes may play key role in plant phenotypes and shaping genome evolution (Supplementary Fig. 7).

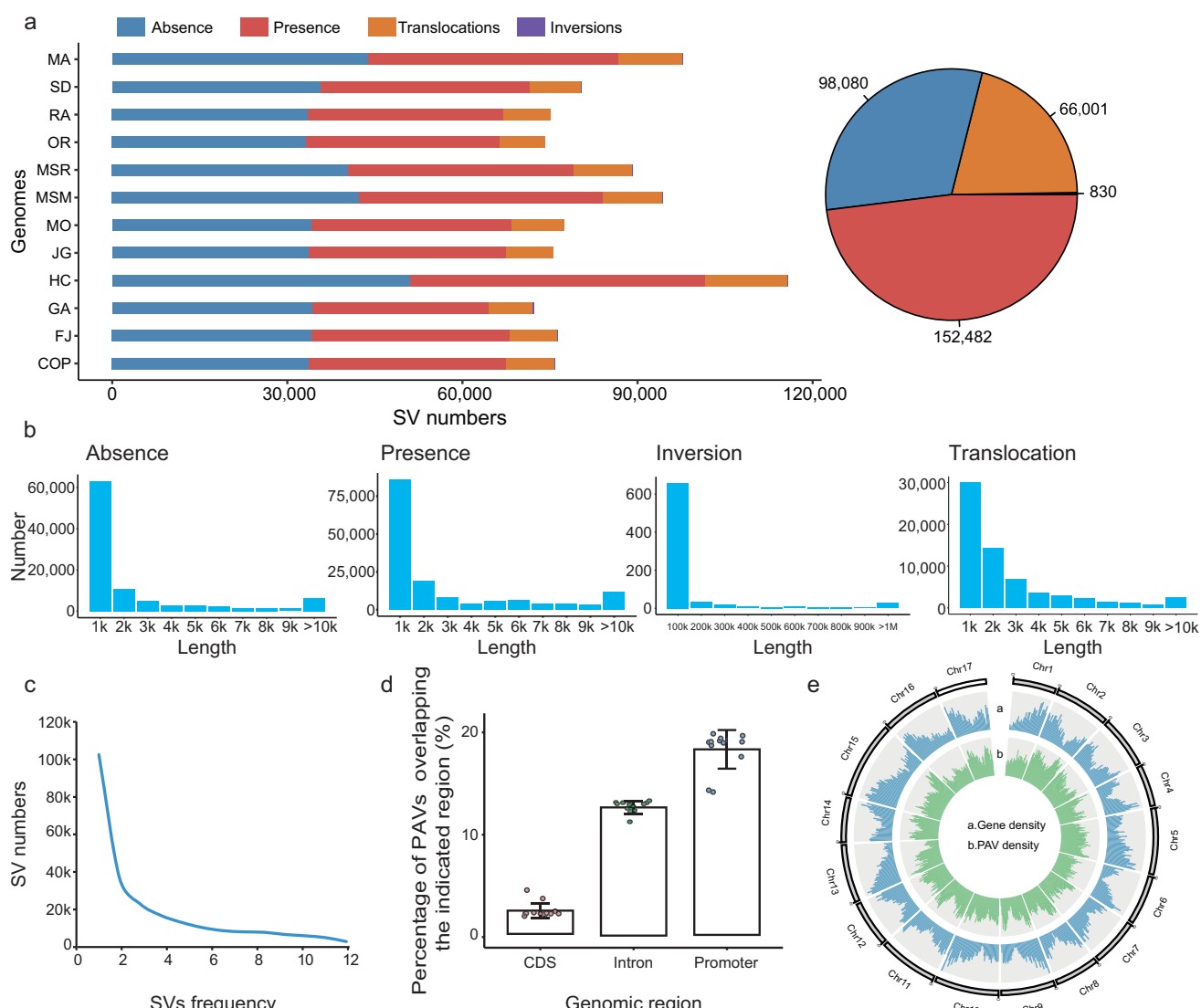

**Fig. 4 | Genetic variations in 12 apple genomes. a** The numbers of four types of structural variations (SVs) in each apple accession. **b** The sequence lengths of the four SV types in the apple accessions. **c** The frequency of SVs in apple accessions indicated that most SVs were present in one, or only a few, accessions.
**d** Percentages of SVs sharing overlap with different genomic regions in the 12 apple accessions. Each point is one accession. Data are presented as mean values ± SD of 12 independent accessions. **e** The PAV distribution across a map of the apple reference genome, demonstrating the number of PAVs per Mb. Source data are provided as a Source Data file.

We also determined SV distribution characteristics, and found that SV frequency decreased with increased input of genome number, which again suggested that many specific SVs were presented in the different cultivars (Fig. 4c). Furthermore, we determined that 71.31% of PAVs were formed by TEs, which LTR/Gypsy TEs were the most abundant, accounting for 43.27% (Supplementary Fig. 8). This result was consistent with previous studies showing that evolution of LTR-RTs has created abundant genetic diversity among species[63]. Most PAVs were present in promoter regions, while the coding sequence (CDS) regions contained the fewest (Fig. 4d). The PAVs were mainly concentrated on chromosome 5 and chromosome 15 (Fig. 4e).

There was an uneven distribution of SVs along the chromosomes, with 123 SV hotspot regions (Supplementary Data 18; Supplementary Figs. 9 and 10), indicating that multiple, independent SVs have arisen in these regions. Chromosome 16 might be the most conserved because all SV numbers of each window along the chromosome were below 100 (Supplementary Figs. 9 and 10). We found that 11 SV hotspot regions on chromosome 8 (Supplementary Data 19), which involved four genes, were clustered in the phenylpropanoid biosynthesis pathway in the first of 11 SV hotspot region by KEGG analysis. We also observed that five genes involved in the brassinosteroid biosynthesis pathway were significantly accumulated in the fourth hotspot region on chromosome 13 (Supplementary Data 20; Supplementary Figs. 9 and 10). Previously, SVs related to phenylpropanoid and brassinosteroid biosynthesis were reported to be involved in plant development, fruit coloration and response to environment[63–65]. These findings are consistent with a previously reported trend that variants harbored within SV hotspots may undergo stronger environmental selection than those in other genome regions[66]. Our delineation of SV hotspot regions will help broaden the understanding of how dynamic genomic variation has contributed to responses to environmental pressures.

## SVs associated with variations in fruit quality

SVs can influence the expression of nearby genes by altering their sequence or copy number, or by changing the composition or position of *cis*-regulatory sequences[67,68]. We investigated this relationship for the comprehensive SV catalog across our apple pan-genome. The results suggested that 57.46% (143,997) of the SVs overlapped with genes or flanking regulatory sequences (±5 kb of coding sequence); 76,495 SVs were associated with promoters; 13,117 SVs were associated with exons; and 54,385 SVs were associated with introns. Furthermore, of the 42,346 annotated genes, 91.95% had at least one SV within 5 kb of the CDS across the 13 genomes, with the majority found in *cis*-regulatory regions (Supplementary Table 12). To assess the potential impacts of SVs on the expression of specific genes, we performed RNA-seq analysis of 22 samples of three tissue/organ types (leaves, fruit peels and fruit flesh) from the 13 accessions that captured 41,499 SV-related genes (Supplementary Fig. 11). This revealed 4480, 1204, and 8619 genes, overlapping with promoter, CDS or intron sequences, respectively, which were significantly associated with altered expression of the corresponding genes in three tissues (Supplementary Fig. 11). The distribution of SVs upstream and downstream from genes indicated that regions closer to genes are more conserved, while regions farther away from genes have increased frequencies of mutations (for the distribution of SV breakpoints for each 10 bp window, the total number of genes is 46,050) (Supplementary Fig. 12).

Our pan-SV genome has the potential to reveal genes and variants underlying quantitative trait variation that would otherwise be masked by hidden genomic complexity. Fruit coloration, an important factor in the product value of apple, is largely determined by the contents of anthocyanins[69]. Many anthocyanin biosynthesis genes, transcription factors (TFs) and related modification genes have been functionally characterized in recent years in apple, but few causal mutations have been identified. We investigated SVs potentially involved in regulating anthocyanin biosynthesis to gain insights into the genetic mechanisms

underlying fruit color. One of these was a 209 bp insertion (LTR/Gypsy TE) in the intron of mitogen-activated protein kinase homolog *MMK2* (GS07G0097300) (Fig. 5a); homologs of MMK2 in *Arabidopsis thaliana* have previously been demonstrated to be upregulated in response to salt and copper stress[70,71]. Whether *MMK2* is involved in fruit development is not known; reverse transcription quantitative PCR analysis (RT-qPCR) indicated that expression of *MMK2* gradually increased during fruit development, and was closely related to anthocyanin accumulation (Supplementary Fig. 13a–c). Interestingly, *MMK2* expression was not detected during fruit ripening in uncolored apples (Supplementary Fig. 12d, e).

A transgenic assay also suggested that major *MMK2* expression promoted anthocyanin accumulation in apple calli, further indicating that *MMK2* functions in apple coloration and *MMK2*-silenced apple fruits exhibited reduced anthocyanin contents relative to the empty vector controls (Supplementary Fig. 13f, g; Fig. 5b–d). To further test the correlation between *MdMMK2* and fruit color, we also conducted a PCR-based analysis in a larger population. We have allelotyped the *MdMMK2* locus in 24 cultivars with varied fruit color phenotypes (https://npgsweb.ars-grin.gov/gringlobal/cropdetail.aspx?type=descriptor&id=115) (Fig. 5e). We confirmed that the cultivars with uncolored fruit were heterozygous for a 209 bp insertion in the intron of *MdMMK2*, while the cultivars with red colored fruit were homozygous without the 209 bp insertion. These results indicate that the *MdMMK2* allele is associated with fruit coloration (Fig. 5e; Supplementary Table 13). A previous study showed that the second intron of *AGAMOUS* (*AG*) encodes several ncRNAs that repress *AG* expression[72], so we speculate that the 209 bp insertion in the fourth intron of *MdMMK2* acts as a ncRNA.

To determine intron mRNA levels, we cloned the fourth intron sequence from apple peel cDNA (Fig. 5f). Interestingly, the fourth intron of *MdMMK2* could be amplified from and was expressed in both colored and uncolored varieties. Subsequently, this region was detected as ncRNA. To avoid genomic DNA contamination, RNAs without RT were interrogated by PCR as a control (Fig. 5f). To explore the effect of sequence variation of *MMK2* on gene expression, we performed a firefly luciferase (LUC) complementation imaging assay in *N. benthamiana* leaves using the fourth intron of *MdMMK2* with insertion (*MMK2-In4*[+]) or without insertion (*MMK2-In4*) as effector, and *MdMMK2* as reporter. Co-expression of the fourth intron with TE insertion of *MdMMK2* and *MdMMK2* repressed *MdMMK2* transcription compared with co-expression of the fourth intron without TE insertion of *MdMMK2* and *MdMMK2* (Fig. 5g–j). Here, we propose that the intronic ncRNAs from *MdMMK2* intron four confers *MdMMK2* expression and may participate in stress-induced anthocyanin accumulation.

There is increasing evidence that many disease resistance-related genes are also involved in regulating plant growth[73–75]. Here, according to the physiological characteristics of fruit, we divided the 13 accessions into different groups, such as disease-resistant and non-disease-resistant; colored and uncolored. We then found SVs related to these characteristics in each group. This revealed nine SVs present in both the disease-resistant related group and the color-related group. The nine shared SVs associated with fruit color and disease resistance, which are potentially involved in trade-offs between fruit quality and immunity, are listed in Supplementary Data 21. Among these, we identified a methyl jasmonate (MeJA) pathway gene *MdMYC2-like* (MD14G1126900) with a 373 bp insertion in its upstream sequence, which may have a regulatory function in fruit coloration and disease resistance. Such data suggest strategies for engineering enhanced disease resistance in apple cultivars without obvious penalties in fruit quality.

## Discussion

In the present study, we combined three existing and ten newly assembled genomes for pan-genome analyses. We identified SVs and

 

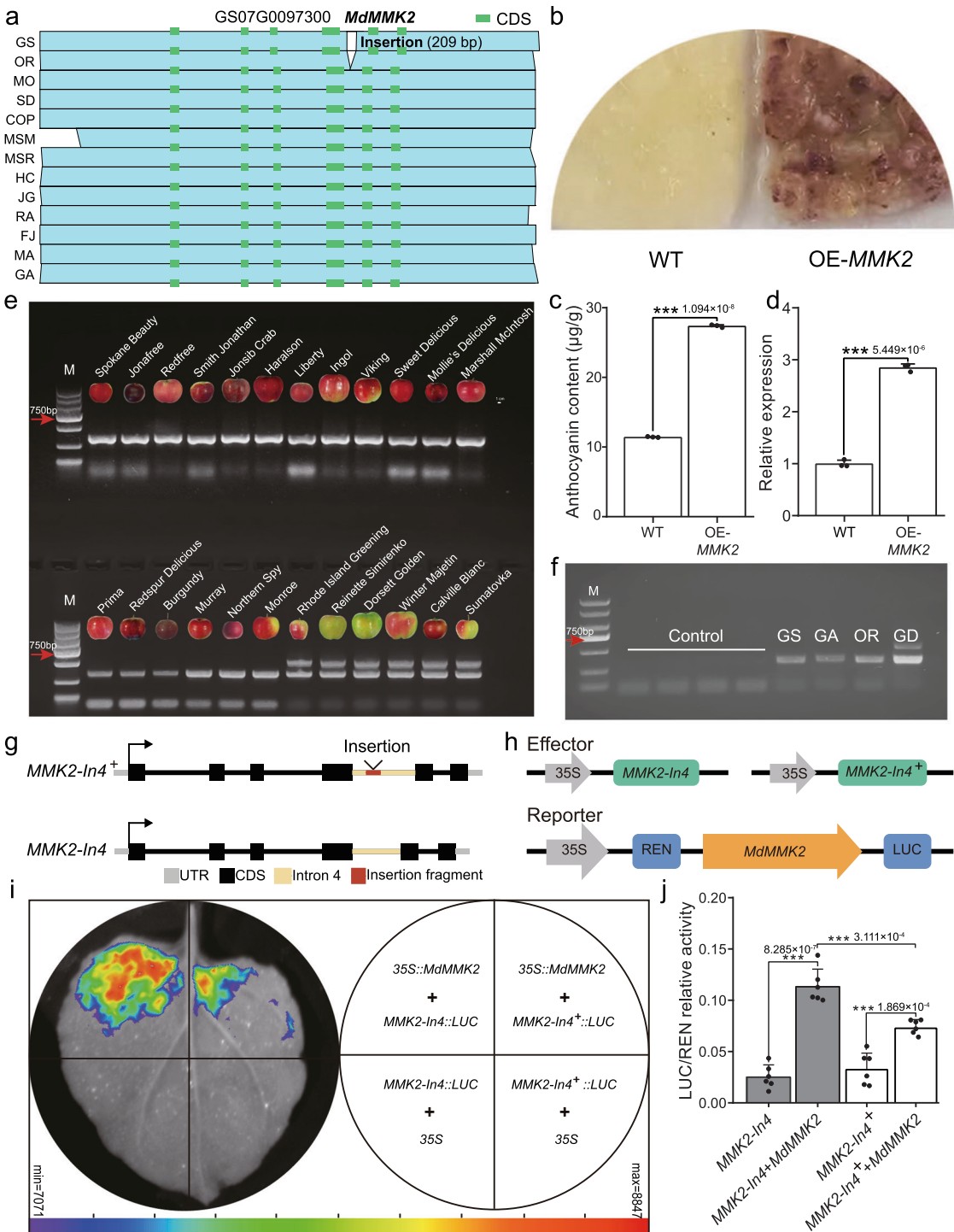

**Fig. 5 | A domestication-related SV potentially associated with fruit color.**
**a** Schematic diagram of SV affecting the *MMK2* gene in colored and uncolored apple accessions. **b** Overexpression of *MdMMK2* in apple calli. **c** Anthocyanin content of transgenic apple calli (µg/g fresh weight). **d** Relative expression levels of *MdMMK2* in apple calli, examined using reverse transcription quantitative PCR analysis (RT-qPCR). RT-qPCR, high performance liquid chromatography (HPLC) and immunoblot analysis were performed using three biological replicates. Data are presented as mean values ± SD of three independent biological replicates. Asterisks indicate significant difference by Student's *t*-test (two-sided; ***p < 0.001). *p*-values are shown in (**c**) and (**d**). **e** Genotype identification of *MdMMK2* in 24 cultivars. M: DNA ladder; Marker 2000. Scale bar 1 cm. The phenotype images are collected from the U.S. Department of Agriculture (USDA), the GRIN database: ([https://npgsweb.ars-grin.gov/gringlobal/cropdetail.aspx?type=descriptor&id=115](https://npgsweb.ars-grin.gov/gringlobal/cropdetail.aspx?type=descriptor&id=115)). Experiments were repeated three times independently with similar results. **f** Identification of the transcription status of the fourth intron in *MdMMK2*. Total RNA (i.e., RNA that was not reverse transcribed) was used as the template in standard PCR reactions as a control. The lack of a band indicated that no genomic DNA remained. The last four lanes show reactions in which apple peel cDNA was used as the template and the primers were specific for the *MdMMK2* fourth intron. Experiments were repeated three times independently with similar results. **g** The schematic of *MdMMK2* fourth intron with insertion (*MMK2-In4⁺*) or without insertion (*MMK2-In4*). **h** Schematic diagram of the effector constructs expressing *MMK2-In4*, *MMK2-In4⁺* and the reporter vector containing the *MdMMK2*. REN, Renilla luciferase; LUC, firefly luciferase. **i** Firefly luciferase (LUC) complementation imaging assay of validation of *MMK2* transcriptional activation **j** LUC/REN ratios analysis of different co-infiltration combinations. Data are presented as mean values ± SD of six independent biological replicates. Asterisks indicate significant difference by Student's *t*-test (two-sided; *** *p* < 0.001). *p* values are shown in (**j**). Source data are provided as a Source Data file.

CNVs associated with fruit coloration, and found that some SVs may also be involved in trade-offs between fruit quality and immunity. Analyzes of the large number of SVs also gained insights into apple evolution and domestication. These genomic resources are valuable for apple genetic improvement through molecular design or genome editing.

Our results are consistent with previous reports that gCNVs contribute to gene expression[13,38]. In *Arabidopsis*, *SAUR72* is induced by auxin[76,77], and functions in regulating cell expansion, leaf senescence, root meristem patterning and auxin transport[59,78]. It is also responsive to dark-to-light transitions[79] and is involved in tolerance of salt stress and drought stress[80]. In our study, we found that the *SAUR72* CNV number correlated with its expression in colored and uncolored apple fruits, and a transient expression assay suggested that *SAUR72* may be an anthocyanin biosynthesis regulator and contributes to apple fruit ripening. Thus, our CNV catalog is useful for investigating hidden genomic variations underlying phenotypic variations. Beyond the effects on quantitatively altering gene expression, many of the gene copies identified in this study likely exhibit neofunctionalization or subfunctionalization. Our previous study showed that following whole-genome duplication, the duplicated *ERF17* gene copies exhibited nonfunctionalization, where one of the copies was pseudogenized[81]. In addition, the apple genome harbors two *MdMPK4* copies due to genome duplication, and that the subfunctionalization of *MdMPK4* copies involved in the regulation of fruit degreening and peel coloration might make fruits more attractive[82,83]. In this study, we identified 20,220 protein-coding CNV genes using CNVnator, which is similar to that reported in rice (25,549 gCNVs)[28]. We also found that the copy number of most CNV genes, which were positively correlated with their expression levels, was less than three, leading us to speculate that any gene with more than three copies may have acquired subfunctions (inducible expression under specific conditions) or have become non-functional (Supplementary Data 15–17).

SVs have been reported to influence important traits such as flavor, fruit size and environmental adaptation[27,84,85]. Our results revealed many protein kinases located at SV hotspot regions. Previous studies showed that the mitogen-activated protein kinase homolog MMK2 functions in stress resistance in plants[70,71]. We found that an SV in *MMK2* is associated with fruit coloration. Since cultivars with improved fruit quality and increased environmental stress tolerance are critical for apple production, further deciphering the role of MMK2 in growth-defense trade-offs holds great promise to achieve the breeding objective. This study illustrates how high-quality long-read genome assemblies can reveal complex causative variants for agronomic traits. In addition, our delineation of SV analysis showed that the MMK2 gene possesses a fruit coloration related insertion in an intron, suggesting that rapid evolution may have driven by adaptation to varied environmental pressures that allowed plants to adjust growth and defense based on external conditions.

## Methods
### Genome sequencing
For *Malus* assembly, MA, MO, COP, FJ, GS, HC, JG, OR, RA, and SD mature fruit were collected from the Northern suburb farm (Beijing, China). Samples were collected from a single plant for each of the ten accessions. For circular consensus sequencing (CCS), genomic DNA was extracted from in vitro grown seedlings using the DNeasy Plant Mini Kit (Qiagen). A 15 kb library was constructed and sequenced using the Pacific Bioscience Sequel II platform (Annoroad Gene Technology). On average, 26.10 Gb CCS reads of each accession with an N50 size of 15.06 kb were generated using ccs software v3.0.0 (https://github.com/pacificbiosciences/unanimity/). For Hi-C library construction, leaf tissue from GS and FJ were vacuum infiltrated with 1% formaldehyde for 15 min to separate the nucleus and then abraded with liquid nitrogen and washed with extraction buffer I (0.4 M sucrose, 10 mM

Tris-HCl, pH = 8.0, 10 mM $MgCl_2$, 5 mM β-mercaptoethanol, 0.1 mM phenylmethylsulfonyl fluoride (PMSF) and protease inhibitor), extraction buffer II (0.25 M sucrose, 10 mM Tris-HCl, pH = 8.0, 10 mM $MgCl_2$, 1% Triton X-100, 5 mM β-mercaptoethanol, 0.1 mM PMSF and protease inhibitor) and extraction buffer III (1.7 M sucrose, 10 mM Tris-HCl, pH = 8.0, 0.15% Triton X-100, 2 mM $MgCl_2$, 5 mM β-mercaptoethanol, 0.1 mM PMSF and protease inhibitor). Digestion with restriction enzyme (*Dpn*II, 400 units of MboI) for chromatin digestion was run overnight at 37 °C on a rocking table[86]. Hi-C libraries were created using an Illumina TruSeq DNA Sample Prep Kit and were sequenced using an Illumina HiSeq X Ten with 2 × 150 bp reads. An average of 111.24 Gb of data was generated for each accession. For RNA-Seq, leaf samples from the 13 cultivars were collected separately after bud germination, and fruit peels and flesh samples from five cultivars (FJ, GS, OR, SD and GA) were collected at 100–180 days after full bloom (DAFB). Total RNA was isolated using the RNAprep Pure Plant Kit (TIANGEN). RNA-seq library construction was performed from extracted samples and libraries were sequenced on the Illumina NovaSeq platform. Each sample yielded more than 6 GB of data with ~30–50 million paired-end 150 bp reads per sample. Clean data were obtained after removing unstable connectors. The clean reads were used for further analysis[87].

Apple callus formation was induced from young embryos of the OR apple cultivar (*M. domestica* Borkh.) and calli were subcultured in MS (Murashig and Skoog) medium containing 0.5 mg/L indole-3-acetic acid (IAA) and 1.5 mg/L 6-benzylaminopurine (6-BA) at 23 °C in the dark. The calli were subcultured three times at 15-day intervals before being used for genetic transformation and other assays[88–90].

### Genome size estimation
Genome size was estimated by *k*-mer frequency analysis. The distribution of *k*-mers depends on the characteristic of the genome and follows a Poisson's distribution. Before assembly, the 17-mer distribution of CCS reads was generated using Jellyfish (v2.2.6)[91].

### Genome assembly
Contig genome assembly of the ten accessions was performed from CCS clean reads using the hifiasm[92] software with parameters settings: "-l 2 --purge-cov 100 --high-het -s 0.01". For GS and FJ de novo genome assembly was performed to anchor contigs into chromosomes using Hi-C reads using HiC-Pro (v2.11.1)[93] and LACHESIS (v2.0)[94]. Finally, the assembled genomes were manually corrected with Juicebox (v1.11.08)[95]. For MO, HC, JG, COP, RA, OR, SD and MA reference-guided genome assembly was performed to anchor contigs into chromosomes. The assembly contigs were aligned to their most closely related reference genome using NUCmer (v3.23)[96]; this anchored the contig set to pseudochromosomes by collinearity. BUSCO[43] and LAI[44] were used to determine the completeness based on the eudicots_odb10 database and full-length long terminal repeat retrotransposons, respectively. Genomic synteny for each alignment was used to build whole-genome synteny between GS and the other nine *Malus* genomes using *MCScanX*[97].

### Genome annotation
The same pipeline was used for genome annotation of ten of the apple genomes. The pipeline for prediction of repeat elements included de novo and homology-based approaches. For homolog evidence and alignment searches were performed using the RepBase database (http://www.girinst.org/repbase), and then RepeatProteinMask (http://www.repeatmasker.org/). For de novo annotation, LTR_FINDER (v2.8.7)[98], PILER[99], RepeatScout (http://www.repeatmasker.org/) and Repeat-Modeler (http://www.repeatmasker.org/RepeatModeler.html) were used to construct de novo libraries, then annotation was carried out with Repeatmasker (http://repeatmasker.org/). We note that several other studies have conducted TE annotations using the same

methods[85,100,101]. The principle of TE annotation mentioned above is similar to the working principle of EDTA[102].

A strategy combining ab initio prediction, protein-based homology searches and RNA sequencing was used to annotate the gene structure. Protein sequences from *Pyrus sorotina*, *Prunus persica*, *Prunus armeniaca* and *Fragaria vesca*, were aligned to the corresponding genome using WUblast (v2.0)[103] with an *E* value cut-off of $1 \times 10^{-5}$ and the hits were conjoined by Solar software (v1.0)[104]. GeneWise (v2.4.1)[105] was used to predict the gene structure of the corresponding genomic regions for each WUblast hit. Gene structures created by GeneWise were denoted the Homo-set (homology-based prediction gene set). Gene models created by PASA (v2.3.3)[106] were denoted the PASA Iso set, and the training data for the ab initio gene prediction programs. Five ab initio gene prediction programs: Augustus (v2.5.5)[107], Genscan (v1.0)[108], Geneid (v1.4)[109], GlimmerHMM (v3.0.1)[110] and SNAP (2013.11.29)[111], were used to predict coding regions in the repeat-masked genome. RNA-seq data (Supplementary Table 8) were mapped to the assembly using Tophat (v2.0.8)[112] and Cufflinks (v2.1.1)[113] and then used to assemble the transcripts into gene models (Cufflinks-set). In addition, gene models were predicted from Trinity-assembled transcripts by PASA, referred to here as the PASA-T-set (PASA Trinity set). Gene model evidence from the Homo-set, PASA-Iso-set, Cufflinks-set, PASA-T-set and ab initio programs were combined by EVidence-Modeler (EVM) (v1.1.1)[114] into a non-redundant set of gene annotations. Weights for each type of evidence were set as follows: PASA-ISO-set > Homo-set > PASA-T-set six > Cufflinks-set > Augustus > GeneID = SNAP = GlimmerHMM = Genscan.

The predicted protein sequences were assigned functions by searching the NR, InterPro, Gene Ontology, KEGG and Swiss-Prot protein/function databases. InterPro, Gene Ontology and KEGG were searched by InterProScan[115] with the following parameters: "-f TSV -dp -goterms -iprlookup -pa". NR, Swiss-Prot and TrEMBL were searched using BLAST with an *E* value cut-off of $1 \times 10^{-5}$. The results from these databases were concatenated. ClusterProfiler[116] was used to perform the Gene Ontology term analysis and KEGG enrichment analysis.

## Construction of the *Malus* family-based pan-genome

For the pan-genome, core and accessory (dispensable) gene sets were estimated based on gene clustering. The OrthoFinder (v1.1.4)[45] clustering results were used to identify gene clusters that were shared by the 13 *Malus* accessions, which we defined as core gene clusters. Gene clusters were considered soft-core cluster that occurred in 11 to 12 accessions. Gene clusters were considered accessory (dispensable) if they occurred in two to ten accessions. Gene clusters that existed in only one genome were defined as accession-specific clusters.

## Gene-based pan-genome construction

We employed a stepwise strategy to build the gene-based pan-genome. First, we carried out pairwise collinearity analysis for the 13 assemblies using *MCscan* (python version) with default parameters. We used GS genes as the base, and these genes from 12 assemblies were added in a stepwise manner: a gene will be added into the gene list and assigned a new locus ID if it has no collinear gene with any genes in the gene list produced by the preceding step. This operation was repeated until all genes from 12 assemblies had been added to the pan-genome.

## Phylogenetic analysis of the genomes

A phylogenetic tree comprising the 13 *Malus* accessions and four outgroup genomes was constructed using 554 single-copy orthologous genes identified by OrthoFinder[45]. Muscle (v3.8.31)[117] was used with default parameters to perform multiple sequence alignment for single-copy orthologous genes. The protein alignment was transformed into codon alignments and then combined to make a super alignment matrix. The phylogenetic tree of the 13 *Malus* accessions

and four outgroup genomes was constructed using RAxML (v8.0.19) with the following parameters: -f ad -N 1000 -m PROTGAMMAAUTO. Phylogenetic tree analysis of the 13 *Malus* accessions was performed using IQ-TREE (v1.6.6), based on the best model (GTR + F + ASC + R7) determined by the Bayesian information criterion. Bootstrap support values were calculated using the ultrafast bootstrap approach (UFboot) with 1000 replicates. Finally, the MCMCtree program implemented in PAML[118] was applied to infer the divergence time with the following parameters: burn-in, 5,000,000; sample-number, 1,000,000; sample-frequency, 50. The calibration times of divergence were obtained from the TimeTree database (http://www.timetree.org/). According to the clustering results, gene clusters with abnormal gene numbers in several species were filtered out and then the expansion and contraction of gene clusters were analyzed using CAFÉ software (v2.1)[119]. Positively selected genes were identified with the branch-site model in the PAML software[120].

## Detection of genomic variations

We aligned the 13 *Malus* genomes to the GS reference genome and then combined three methods to identify SVs: Assemblytics[61] (http://qb.cshl.edu/assemblytics/), SyRI[60] (https://github.com/schneebergerlab/syri) and SVMU[62] (https://github.com/mahulchak/svmu). We extracted alignment pairs from any pair of genomes based on NUCmer (v3.23) (--mum --maxgap = 500 --mincluster = 1000) to serve as input for the packages of these three software programs with default parameters. Using the above pipeline, we obtained three raw SV sets. We then merged absence and presence similarly to a previous study[28]: two SVs were merged into one if the overlapping ratio of these two SVs (the length of the overlapping sequence/ the length of non-redundant genome segment covered by the two SVs) exceeded 90%. For inversions and translocations, we only considered candidates supported by SyRI. We combined SVs (including absence, inversion and translocation) relative to *Malus* from the 13 accessions, with 90% similarity as the threshold. For the absent SVs, if the overlapping ratio of two absent SVs (the length of the overlapping sequence/the length of non-redundant genome segment covered by the two SVs) exceeded 90%, we determined that these two absent SVs from two callsets were the same absent SV. For present SVs from the 13 accessions, if their overlapping ratio (the length of the shorter SV/the length of the longer SV) exceeded 90%, and their positions on the GS genome were less than 5 bp, the two present SVs were merged into one. To determine whether SVs in the different regions were differentially expressed (|Fold Change| ≥2 and FDR < 0.01), DESeq2 was used, in which the difference test distribution model is a negative binomial distribution[117]. For CNV, the PacBio reads of the 13 accessions were individually aligned to the GS genome, using Minimap2 (v2.24)[121]. Based on the alignments, CNVs were called using CNVnator (v0.4.1)[122]. To analyze gene copy number variations, we aligned the proteins in the CNV regions of GS to each genome assembly using BLAT (v36)[123]. The alignment results were then filtered with a relatively strict cut-off: identity >90% and coverage >90%. Next, the number of loci for which each gene could be mapped to each genome assembly was counted as the copy number of each gene in the corresponding accession.

We calculated the distribution of SV breakpoints for each 400 kb window (with a 200 kb step size) along each chromosome. All 400 kb windows were then ranked in descending order according to the numbers of SVs within the window. We defined the top 10% of all windows with the highest frequency of SV breakpoints as SV hotspots, then merged all of the continuous hotspot windows into "hotspot regions'.

## Transcriptome analysis

Total RNA was isolated from two biological replicates of a sampled organ at a specific developmental stage to investigate expression. As above, RNA-seq libraries were constructed and sequenced on NovaSeq

platform. The clean reads were mapped against the GS genome using Hisat2 (v2.0.5)[124] software. The number of reads mapped was counted using HTSeq (v0.6.1)[125] and then fragments per kilobase of exon per million mapped fragments values were calculated for each gene. Transcripts with fewer than one per million mapped reads were ignored. Analysis of differential gene expression between two samples was performed using the DESeq2 R package (v1.20.0)[126]. Genes with an adjusted $p$ value < 0.05 found by DESeq2 were defined as differentially expressed.

## HPLC analysis

Frozen apple fruit peel and calli samples (0.8–1.0 g fresh weight) were ground in 10 mL extraction solution (methanol:water:formic acid:trifluoroacetic acid = 70:27:2:1) and incubated at 4 °C in the dark for 72 h, with shaking every 6 h. The supernatant was passed through filter paper and then through a 0.22 μm Millipore™ filter (Billerica, MA, USA). For HPLC analysis, trifluoroacetic acid: formic acid: water (0.1: 2: 97.9) was used as mobile phase A and trifluoroacetic acid: formic acid: acetonitrile: water (0.1: 2: 48: 49.9) was used as mobile phase B. The gradients used were as follows: 0 min, 30% B; 10 min, 40% B; 50 min, 55% B; 70 min, 60% B; 30 min, 80% B. Anthocyanins were quantified by measuring absorbance at 520 nm using a calibration curve based on a commercial standard of cyanidin-3-$O$-glucoside (Sigma, St Louis, Missouri, USA). Anthocyanins concentrations were expressed as μg/g of fresh weight (FW)[127]. All samples were analyzed in biological triplicates (extracted from three different apple peels or calli).

## Flow cytometric measurement

Flow cytometry was first conducted to estimate genome size. FJ and GS leaves were collected and analyzed using a CyFlow Space Flow Cytometer (Sysmex Europe GmbH, Norderstedt, Germany), equipped with a UV-LED source (with emission at 365 nm) and a blue solid-state laser (k = 455 nm). GA (2n = 2x = 24) was used as the reference. The genome size was estimated by $k$-mer frequency analysis. The distribution of $k$-mers depends on the characteristic of the genome and follows a Poisson's distribution. Before assembly, the 17-mer distribution of CCS reads was generated using Jellyfish (v2.2.6)[128].

## RNA extraction and RT-qPCR

Total RNA was extracted from apple peel and calli samples using an RNA Extraction Kit (Aidlab, Beijing, China) according to the manufacturer's instructions. DNase I (TaKaRa, Ohtsu, Japan) was added to remove genomic DNA and the samples were converted to cDNA using the Access RT-qPCR System (Promega, Madison, WI, USA), according to the manufacturer's instructions. Gene expression levels were analyzed by RT-qPCR using 2 × SYBR Green qPCR Mix (TaKaRa, Ohtsu, Japan) on a Bio-Rad CFX96 Real-Time PCR System (BIO-RAD, Hercules, CA, USA), according to the manufacturer's instructions. PCR primers were designed using NCBI Primer BLAST (https://www.ncbi.nlm.nih.gov/tools/primer-blast/). RT-qPCR analysis was carried out in a total volume of 20 μL, containing 10 μL of 2× SYBR Green qPCR Mix, 0.1 μM specific primers (each), and 100 ng of template cDNA. The reaction mixtures were heated to 95 °C for 30 s, followed by 39 cycles at 95 °C for 10 s, 50–59 °C for 15 s, and 72 °C for 30 s. A melting curve was generated for each sample at the end of each run to ensure the purity of the amplified products. $MdActin$ (LOC103453508) was used as the internal control. The data were analyzed using the internal control and the $2^{\wedge(-\Delta\Delta Ct)}$ method[129]. All primer sequences are listed in Supplementary Data 22. Three biological replicates of the fruit and calli samples were analyzed.

## Construction of expression vectors and stable transformation of apple calli

Overexpression vectors were made by amplifying the full-length $MdMMK2$ sequence from "Red Fuji" apple peel cDNA using RT-qPCR.

To fuse the $MdMMK2$ coding sequence with eGFP (enhanced green fluorescent protein), the corresponding cDNA was cloned into the pGFPGUSPlus plant transformation vector[130] downstream of the CaMV 35 S promoter. All primers used are listed in Supplementary Data 22. For transforming "Orin" calli, resuspension solution (4.43 g/L MS + 30 g/L sucrose +1.5 mg/L 2,4-D + 0.4 mg/L 6-BA + 0.1 μM AS, pH = 6.0) was made, and calli that had been growing well for 3 weeks were added and cultivated with oscillation at 28 °C and 200 rpm for 20 min. The resuspension solution was aspirated and apple calli were taken out, surface moisture was blotted with sterile filter paper and spread onto the calli co-culture medium (4.43 g/L MS + 30 g/L sucrose + 8 g/L agar + 1.5 mg/L 2,4-D + 0.4 mg/L 6-BA + 0.1 μM acetosyringone (AS), pH = 6.0) in the dark for 2 days. The calli were transferred into the screening medium (4.43 g/L MS + 30 g/L sucrose + 8 g/L agar + 1.5 mg/L 2,4-D + 0.4 mg/L 6-BA + kanamycin 30 mg/L + cefotaxime sodium 250 mg/L, pH = 6.0) and incubated in the dark for about 3 weeks. Wild-type calli were used as a control[131]. Three independent transgenic apple callus lines were obtained for subsequent experiments.

## Agrobacterium-mediated transient transformation

Bagged "Gala" apple fruit were harvested 90 DAFB for transient expression studies. A partial sequence of $MdSAUR72$ and $MdMMK2$ was integrated into the TRV2 vector to generate the TRV2-$MdSAUR72$ and TRV2-$MdMMK2$ constructs[132]. The recombinant plasmids were transformed into $Agrobacterium\ tumefaciens$ strain GV3101 and integration carriers were collected by centrifugation at 5000 rpm for 10 min and resuspended in infiltration buffer (10 mM MES, 10 mM MgCl$_2$, 200 μM AS, pH = 5.6). $A.\ tumefaciens$ lines containing the constructs were transformed into "Gala" fruit[132–134]. The infected apples ("Gala") were placed at 23 °C for either 2 days (for overexpression) or 5 days (for silencing). Infiltration was repeated using three biological replicates.

## Dual-luciferase reporter assay

$MdMMK2$ were amplified by PCR and cloned into the pGreenII-0800-LUC vector as reporters to generate the reporter constructs[135]. The fourth intron of $MdMMK2$ with insertion ($MMK2$-$In4^+$) or without insertion ($MMK2$-$In4$) was used as an effector and cloned into the pGreenII-62-SK vector[136], which were co-infiltrated with reporter into $N.\ benthamiana$ leaves by $Agrobacterium$-mediated transient infiltration. The pGreenII 0800-LUC vector and $MMK2$-$In4$ or $MMK2$-$In4^+$ were co-infiltrated as the control. After 12 h of dark cultivation, samples were transferred to light cultivation for 2 days infiltration. The infiltrated areas of $N.\ benthamiana$ leaves were then ground to a fine powder in liquid nitrogen and resuspended in 1× Passive Lysis Buffer. The LUC/REN ($Renilla$ luciferase gene) ratios were calculated using the Dual-Luciferase Reporter Assay System according to the manufacturer's instructions (Promega, Madison, USA) on a Glo-Max 20/20 luminometer (Promega). LUC/REN relative activity was repeated using six biological replicates.

## Statistical analysis

Experimental data were analyzed by two statistical methods. For comparison of two groups, we used Student's $t$-test, two-sided, *, $p < 0.05$; **, $p < 0.01$; ***, $p < 0.001$. For multiple comparisons, ANOVA was used followed by Duncan's test, as mentioned in the figure legends. GraphPad Prism 8.0, Microsoft Excel 2016 and IBM SPSS Statistics 22 were used for analysis.

## Reporting summary

Further information on research design is available in the Nature Portfolio Reporting Summary linked to this article.

## Data availability

All RNA-seq data generated in this study are available at the National Center for Biotechnology Information (NCBI) BioProject database

under accession PRJNA872768. The genome sequences of COP, FJ, HC, JG, OR, RA, SD, MO and MA are accessible under NCBI BioProject PRJNA869488. The genome sequence of GS is available under NCBI BioProject PRJNA927238. The gene cluster data [https://figshare.com/s/4e1ea61459393ff54684], the PAV data [https://figshare.com/s/c6734dc864c07f51df9c], and the GFF3 (General Feature Format Version 3) files (annotation of 10 assemblies) [https://figshare.com/s/5fea32dd0fba11ead6bd] are all available at figshare. Please note that *M. asiatica* "Zisai Pearl" also abbreviated as "ZP" in the description of the deposited data i.e., "MA" and "ZP" represent the same plant material. Source data are provided with this paper.

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

## Acknowledgements

The work was supported by the National Key R&D Program of China (2022YFD2100100 to T.Wu), the National Natural Science Foundation of China (32322074, 32072543 to T.Wu), the 111 Project (B17043 to T.Wu), and the 2115 Talent Development Program of China Agricultural University (to T.Wu). We thank Dr. Fei Shen for providing assistance with data analysis, Dr. Kenong Xu for providing germplasm resource DNA samples, and Dr. Ke Cao for critical reading and comments on the manuscript.

## Author contributions

T.Wu conceived and designed the research. T.Wang, S.D., and C.X. conducted the experiments. Y.W., X.Z., L.C., X.X., and S.D. contributed reagents and analytical tools. Z.H. gave advice and edited the manuscript. T.Wang and S.D. wrote the manuscript. All authors read and approved the manuscript.

## Competing interests

The authors declare no competing interests.
