## [Peer Review File · Nature Communications]

Pan-genome analysis of 13 Malus accessions reveals structural and sequence variations associated with fruit traitsReviewers' Comments:

Reviewer #1:

Remarks to the Author:

Wang et al describe the sequencing and assembly of genomes from multiple accessions of apple, both cultivated and uncultivated. They have used high-throughput bioinformatic tools to compare the genomes and present some analysis of features of the apple genomes.

The work provides a valuable resource for researchers studying apple including both the commercialisation and the domestication. As a major domesticated tree crop apple is an interesting research subject and having access to these annotated and assembled genomes is useful. The work expands on the previous apple genome sequences and in particular the sequence of non-cultivated accessions is extremely valuable.

However as noted below I have significant concerns about some of analyses applied to the genome data. I believe that in several cases the authors have over-interpreted the data in an effort to provide specific conclusions. The quality of the supplementary tables is poor with many consisting of a spreadsheet of numbers with no interpretation of explanation and no way to connect numbers to specific genes or traits. Overall there is a lack of connection to phenotype. Which would have made the work significantly more valuable to the field.

I have several specific concerns detailed below

Line 122 The wording makes it sound as if all three tissue types were sampled for all 13 accessions. The legends for the supplementary figures are insufficient to interpret the figures and need to be expanded. For example the S5 legend says "KEGG pathway enrichment analysis of apple fruit color-related genes." Yet the text describes the figure as showing a comparison between accessions with coloured and uncoloured fruit

I found the "dispensable genes" category somewhat unhelpful as it lumped together genes that were present in 12 of the 13 accessions with genes that were present in only 2 of the 13 accessions. This made statements such as line 144 "Notably, dispensable genes were significantly enriched in ..." less compelling. The legend for Fig.S4 is not helpful and the figure appears to show only core genes not dispensable genes and implied in the text. A category for genes present only in domesticated accessions would have been interesting.

I would have preferred to see an unrooted phylogenetic tree and I'm not certain assigning timepoints to the nodes is particularly informative, although this appears to have been done in order to use the CAFE tool. There is considerable historical data covering apple breeding and some comparison of the high-throughput phylogenetic analysis with the known breeding history of the accessions would have been useful. There is no measure of support for nodes shown in either tree and there appears to be a difference in the structure of the two trees with respect to MA, MO, MSR and MSM with these four accessions having different relationships between the two trees. Again this second tree should be unrooted and it is not clear what outgroup was used to place the root shown nor is it clear how the second tree was generated.

Table S12 is just a list of numbers and provides no information of use to the reader

I am not sure what is meant in line 173 by "...such that 183 and 6 families in Malus, and 190 and 3 families in Pyrus had expanded ..." I presume it refers to the output of CAFÉ which predicts 183 expanded families and 6 contracted families for the node where malus diverges. However, it is not clear if this informative and it is my understanding of the software that a statistical significance to these numbers should also be present. Without any discussion of specific gene families that have been gained or lost it is very difficult to assess the value of this data. I was hoping the supplementary tables might have included a list of families that had been gained/lost.

Line 176 the sentence while correct seems irrelevant. This paragraph appears to claim some positive selection (K_a/K_s) of stress response genes but the data is not presented. Table S13 appears to simply list the number of positively selected gene families, but there is no detail and no comparison of specific gene families shown, neither the legends nor the methods explain what was compared. The authors identify 20220 genes with CNVs. Table S14 appears to show many (most?) of these are the presence of a gene in one accession and absence in the other 12 accessions or perhaps presence in one other accession. I'm not sure this is what is normally understood as a CNV although there does

not appear to be any strict definition in the field. I think most readers would find the observation of nearly half the genes in 13 closely related accessions defined as subject to CNV as surprising. A clear definition of what the authors are calling a CNV would be helpful to put this number into context especially since the majority of CNVs identified appeared to have no effect on expression in leaves or fruit. Are there any features of CNVs associated with expression changes that are unique and might provide some predictive value as to whether a given CNV was worth investigating further?

There is no method given for the transient expression studies shown in Figure 3F nor any description of the vectors used

Line 215 Again Table S19 is poorly described and it is unclear to me what is meant by "PAV length was significantly longer..." Figure 4B shows inversions as being longest and translocations longer as well although the total length of all PAV SVs combined is longer.

I could not understand what was meant by Line 217. I think the authors are referring to Figure 4C which shows most SVs are present in only one accession I would not normally expect this to be described as allele frequency.

Line 220 The statement that PAVs were concentrated on two particular chromosomes is not supported by figure 4E which shows the distribution to be relatively even. I accept that there is an uneven distribution of SVs along the chromosomes (hot spots) but not between the chromosomes.

The paragraph starting at line 222 appears to conclude that because some known genes are found in regions with SVs that they are undergoing selection without any supporting data for this conclusion.

The paragraph starting at line 235 suggests a relationship between SVs and genes that results in altered expression. However, there is no definition of "altered expression". Figure S7 relates SVs in different regions to altered expression but it is not clear whether it is the position of the SV or the expression that is significant nor what measure of significance was used. This section needs to be clarified and expression data and statistical tests included.

The paragraph starting at line 265 talks about "shared SVs" it is not clear what is meant by this term. It is not clear why these nine genes have been selected from the 46k genes in the genome or how the SV is connected.

In the discussion line 277 claims the authors identified selective SVs and CNVs underlying several important traits yet this data is not present in the manuscript.

Reviewer #2:

Remarks to the Author:

In this manuscript Wang et al. report on the de novo assembly of 10 apple genomes (wild and cultivated ones) one of which at gold standard level. They then take advantage of these assemblies to look for gene copy number variations between the different apple accessions. The authors then find a correlation between gene copy number and expression levels. Using a candidate approach the authors then show that MdSAUR72 may play a role in apple fruit coloration. Similarly, they could associate MdMMK2 with fruit coloration.

Overall, the manuscript is clear and quite straightforward, but I have some reservations and questions that must be clarified:

Please explain in more detail how exactly the different wild and cultivated apple accessions have been selected. Were there any reasons from a genetic diversity level to select those accessions?

The approach for transposable element annotation is non-standard. Why did the authors choose this approach and not some standard pipeline such as REPET or EDTA?

In line 134 the authors present a range of core genes. Should that not be a single number as these genes must be in common between all accessions?

Why have the authors only focused on gene CNV and not also transposable element CNVs? These could be very interesting too, particularly since the authors often observe CNVs in promoters. I suspect that this could be the result of a TE CNV (such as a MITE).

Concerning the insertion at MdMMK2: what is the sequence there, is it a TE?. Also, to strengthen the correlation with fruit color, it must be tested in a larger population whether this insert is present or not

and if that influences fruit color (e.g., by PCR).

In the discussion the authors state that: "We identified selective SVs and CNVs underlying fruit size, flesh firmness, acidity, stress resistance, and found that selective SVs may also involve trade-offs between fruit quality and immunity." This statement should be toned down as the results presented here are purely correlative and this statement quite speculative.

Finally, how were the copy numbers correlated with gene expression? How were the different haplotypes separated or were all reads mapped to one copy of the corresponding gene? Did the authors observe that all additional copies were expressed or were some individual copies more highly expressed?

Minor comments:

Line 71 please define gCNVs

Line 111 does not make sense to me, does the length correspond to the sum of all TE lengths added together? Please rephrase.

Line 122 is a repetition of what is already mentioned in line 114 and 115.

Line 299 incomplete sentence

Reviewer #3:

Remarks to the Author:

The paper by Wang et al describes how they built a pangenome by integrating existing Malus genomes with newly sequenced ones, obtained with long reads. Only the Granny Smith (GS) obtained the gold standard label according to LTR integrity, with the others achieving close scores. They carry out a comprehensive analysis of the pangenome which I think can be inspiring for other people working on fruit trees. There 's room for improvement though (see my comments below), and my first request would be to make the pangenomic resources produced for this paper available to everyone. That would be the most important single feature that will increase the impact of this work.

Comments

1. Please scan and correct spelling throughout the text, for instance at L21, L64, L225.

2. Sentences L15, L67 and L778 are apparently contradictory, please clarify how many new assemblies were produced. After reading page 4 it seems a total of 10 genomes were assembled.

3. The BUSCO scores on L95 and L104 are also in contradiction. How can core genes achieve higher BUSCO completeness (98.8 > 98.4)?

4. I guess you refer to non-redundant TE sequences in L118, but please clarify.

5. L135 why core estimates are ranges instead of a single number? Please explain.

5.1 Figure 1A: please justify why families are used instead the usual gene clusters (as those in L134). In my opinion a family-based analysis assumes that all each family corresponds roughly to a biological function, which is clearly an over-simplification. In fact see L293.

5.2 Figure 1C: was the genome order shuffled to produce this figure? That's how Tettelin et al did it to avoid implicit biases.

5.3 Figure 1C: a Tettelin-like function would allow you to estimate the core size in the limit; indeed the data with 13 genomes suggests the core will shrink a bit more if more genomes are added (compare this to L137).

5.4 Please make the pangenome clusters available for download so that readers can use them straight away.

6. In the context of L131-132 please check the papers by Michele Morgante where term dispensable is

discussed. I know this is only terminology but I advocate for the term accessory instead. Other people use the 'shell' term.

7. I believe the section from L151 needs refocus. What I mean is that the phylogeny all the way up to Arabidopsis is probably not that relevant for this paper. Instead, the new data presented here would allow to make a high-resolution phylogeny of the Malus genus and the cultivars. Of course that would require trees with bootstrap or equivalent branch support scores, which are not currently shown.

8. I feel the same about the CAFE analysis of family expansions. I believe is of little interest as currently shown.

9. Please explain what you mean with unbiased alignment on L188.

10. Please explain how similar two genes have to be considered 'CNV genes'. I guess on Figure 3 the expression values of all copies are added up? Are the same primers used for the qRT-PCR experiments?

11. Have the authors cross-checked whether the PAVs overlapping genes are confirmed in the OrthoFinder gene clusters? That would allow them to do a internal quality control. This would work both ways: gene clusters might lend support to called PAVs, and the otehr way around. This would also raise the standard of this study and will make the most of the assemblies at hand.

12. On L228 'enriched' probably means 'significantly accumulate', am I right?

13. Can you please compute the numbers of PAVs in promoters with different distance thresholds? There are published reports that suggest that promoters in other fruit crops (peach, for instance) are mostly conserved up to 500bp. Of course there are documented cases of conserved regulatory sequences several Kb away, but the genome-wide trend is that proximal promoters are much shorter than 5kb. In fact, the RNAseq expression data would allow to estimate this with some precision, by looking only at genes with altered expression patterns (L246).

14. Please make the PAVs available to readers, this will accelerate apple research and will increase the profile of this paper.

15. Please rephrase L256, as that kinase has been linked to salt and copper stress in *A. thaliana*, not in *Malus*, where it could have different functions. In fact, the whole section about MMK2 is quite interesting but unfortunately no mechanistic model is provided to link the expression of this protein and anthocyanin content. Figure 5 shows that MMK2 could be an excellent marker for anthocyanin content nevertheless.

16. L351 The description of the method of genome size estimation is clearly insufficient, more details are needed if reader are to be able to reproduce this.

17. L365 Why was embryophita used instead of eudicots for BUSCO analysis? Surely the latter makes more sense for *Malus*.

18. FigureS2B: please indicate how the plot was produced.

19. L458 Please indicate where readers can obtain the GFF files and how isoforms of the same gene were treated

20. Please correct reference 43 in L98

Response to Reviewer 1

Reviewer's comment: *Wang et al describe the sequencing and assembly of genomes from multiple accessions of apple, both cultivated and uncultivated. They have used high-throughput bioinformatic tools to compare the genomes and present some analysis of features of the apple genomes. The work provides a valuable resource for researchers studying apple including both the commercialisation and the domestication. As a major domesticated tree crop apple is an interesting research subject and having access to these annotated and assembled genomes is useful. The work expands on the previous apple genome sequences and in particular the sequence of non-cultivated accessions is extremely valuable. However as noted below I have significant concerns about some of analyses applied to the genome data. I believe that in several cases the authors have over-interpreted the data in an effort to provide specific conclusions. The quality of the supplementary tables is poor with many consisting of a spreadsheet of numbers with no interpretation of explanation and no way to connect numbers to specific genes or traits. Overall there is a lack of connection to phenotype. Which would have made the work significantly more valuable to the field.*

Author's response: Thank you for the helpful suggestions. We have now performed additional data analysis and revised related text to make the conclusion more accurate. We have now provided the detail information in supplemental tables and revised the related tables to make supplementary information more clearly. For the connection between the genome data and phenotype, we added the phenotype and detailed information of the 13 *Malus* accessions to describe the genetic diversity between them, we categorized genes into those only present in wild-type apples and those only present in cultivated apples and then conducted the KEGG analysis, we also performed the virus-induced gene silencing (VIGS) of *MdMMK2* in apple fruit to demonstrate causal effect of *MMK2* and we proved that the 209 bp insertion in the fourth intron of *MdMMK2* which may acts as a noncoding RNA to regulate the *MMK2* expression as well as fruit coloration.

Major concern

1. Reviewer's comment: *Line 122 The wording makes it sound as if all three tissue types were sampled for all 13 accessions.*

Author's response: We revised the sentences to: 'We collected leaf samples from each accession, and fruit peel and fruit flesh samples from the five apple accessions (GS, OR, SD, FJ and GA), for a total of 22 samples. RNA sequencing was performed of all 22 samples with a mean of 6 Gb pairs (Table S10)'.

2. Reviewer's comment: *The legends for the supplementary figures are insufficient to interpret the figures and need to be expanded. For example the S5 legend says "KEGG pathway enrichment analysis of apple fruit color-related genes." Yet the text describes the figure as showing a comparison between accessions with coloured and uncoloured fruit.*

Author's response: Thank you for the helpful suggestion. We re-wrote the legends for the supplemental figures (Figure S1, S2, S4, S5) to give sufficient information for interpretation. We also revised the legend of Figure S6 to: 'KEGG pathway enrichment analysis of the up-regulated genes in colored apple fruit compared with uncolored apple fruit'.

3. Reviewer's comment: *I found the "dispensable genes" category somewhat unhelpful as it lumped together genes that were present in 12 of the 13 accessions with genes that were present in only 2 of the 13 accessions. This made statements such as line 144 "Notably, dispensable genes were significantly enriched in ..." less compelling. The legend for Fig.S4 is not helpful and the figure appears to show only core genes not dispensable genes and implied in the text. A category for genes present only in domesticated accessions would have been interesting.*

Author's response: We thank the reviewer for pointing this out. To better categorize the dispensable genes and according to gene classification standard in the reference Liu et al., 2020²⁶, we have divided gene classification into four categories: core, softcore, dispensable and specific genes. We added related sentences in the MS: 'Through gene cluster analysis by OrthoFinder⁴⁵, 14,896 clusters that were present in all 13 accessions were defined as 'core', 6,314-7,408 clusters that occurred in 11 to 12 accessions were defined as 'softcore', 8,226-12,762 clusters that occurred in 2 to 10 accessions were defined as 'accessory (dispensable)', and 766-2,807 clusters that were represented in only one accession were defined as 'specific' (Figure 1C; Table S11)'.

We also re-constructed the KEGG analysis from Figure S4 for the dispensable genes, core genes, softcore genes and specific genes, and revised the legend for Figure S4 to: 'KEGG (Kyoto

Encyclopaedia of Genes and Genomes) pathway enrichment analysis of core genes, softcore genes, accessory (dispensable) genes and specific genes from the 13 apple accessions.’.

To improve our understanding of the genetic differences between cultivated and wild apple species, we categorized genes into those only present in wild-type apples and those only present in cultivated apples. KEGG analysis was then conducted, and the results showed that two secondary metabolism pathways: phenylpropanoid biosynthesis and terpenoid biosynthesis (including diterpenoid biosynthesis, sesquiterpenoid and triterpenoid biosynthesis), as well as two stress resistance pathways; ascorbate and aldarate metabolism, and cysteine and methionine metabolism, were significantly enriched in wild-type apple⁴⁸⁻⁵². These findings highlight a remarkable difference in content and composition of secondary metabolites between wild and cultivated apples fruit also during stress resistance. We added related sentences in the MS.

4. Reviewer’s comment: *I would have preferred to see an unrooted phylogenetic tree and I’m not certain assigning timepoints to the nodes is particularly informative, although this appears to have been done in order to use the CAFE tool. There is considerable historical data covering apple breeding and some comparison of the high-throughput phylogenetic analysis with the known breeding history of the accessions would have been useful. There is no measure of support for nodes shown in either tree and there appears to be a difference in the structure of the two trees with respect to MA, MO, MSR and MSM with these four accessions having different relationships between the two trees. Again this second tree should be unrooted and it is not clear what outgroup was used to place the root shown nor is it clear how the second tree was generated.*

Author’s response: Thank you for raising this issue. To increase the accuracy of the phylogenetic tree, we constructed an unrooted tree, and the relationship between MA, MO, MSR and MSM is now consistent between the two trees. The method was added to the M & M section as: ‘A phylogenetic tree comprising the 13 *Malus* accessions and 4 outgroup genomes was constructed using 554 single-copy orthologous genes identified by OrthoFinder⁴¹. Muscle (v3.8.31)¹⁴ was used with default parameters to perform multiple sequence alignments for single-copy orthologous genes. The protein alignment was transformed into codon alignments and then combined to make a super alignment matrix. The phylogenetic tree of the 13 *Malus* accessions and 4 outgroup genomes was constructed using RAxML (v8.0.19) with the following parameters:

-f ad -N 1000 -m PROTGAMMAAUTO. Phylogenetic tree analysis of the 13 *Malus* accessions was performed using IQ-TREE (v1.6.6), based on the best model (GTR+F+ASC+R7) determined by the Bayesian information criterion. Bootstrap support values were calculated using the ultrafast bootstrap approach (UFboot) with 1,000 replicates. Finally, the MCMCtree program implemented in PAML¹¹⁵ was applied to infer the divergence time with the following parameters: burn-in: 5,000,000, sample-number: 1,000,000, sample-frequency: 50. The calibration times of divergence were obtained from the TimeTree database (<http://www.timetree.org/>). According to the clustering results, gene clusters with abnormal gene numbers in several species were filtered out and then the expansion and contraction of gene clusters was analyzed using the CAFÉ software (v2.1)¹¹⁶. Positively selected genes were identified with the branch-site model in the PAML software¹¹⁵. We also revised the related sentences explaining the apple breeding history to: ‘MA originated in China and clustered into a clearly separated monophyletic clade, with OR and GS with uncolored fruit being closely related (Figure 1B). FJ is a hybrid progeny of ‘RA × ‘Delicious’, while SD is a bud sport of ‘Delicious’, and this was also seen in the evolutionary relationship (Figure 2A). The divergence time between apple and other species was consistent with previous studies, supporting the data of this study^{53, 54}.’.

To clarify the distinctions between these three trees, we revised the figure legend for Figure 2 to: ‘Population structure and genetic diversity of apple accessions. (A) Phylogenetic tree of 13 *Malus* accessions and 4 outgroup accessions. Black characters are estimated divergence times (Ma) based on single-copy orthologous group inference. (B) Unrooted phylogenetic tree of 13 *Malus* accessions. Blue characters are estimated bootstrap values. (C) The expansion and contraction of gene clusters between 13 *Malus* accessions and 4 outgroup accessions. Red characters are the number of expanding gene clusters, green characters are the number of contracting gene clusters.’.

5. Reviewer’s comment: *Table S12 is just a list of numbers and provides no information of use to the reader.*

Author’s response: We deleted Table S12.

6. Reviewer’s comment: *I am not sure what is meant in line 173 by “...such that 183 and*

6 families in Malus, and 190 and 3 families in Pyrus had expanded ...” I presume it refers to the output of CAFÉ which predicts 183 expanded families and 6 contracted families for the node where malus diverges. However, it is not clear if this is informative and it is my understanding of the software that a statistical significance to these numbers should also be present. Without any discussion of specific gene families that have been gained or lost it is very difficult to assess the value of this data. I was hoping the supplementary tables might have included a list of families that had been gained/lost.

Author’s response: Thank you for highlighting a potential area of concern. We conducted an enrichment analysis of expanded and contracted genes in all cultivated apples and added the related text in the MS: ‘To improve the understanding of the significance of expanded and contracted gene clusters, we conducted KEGG analysis of expanded and contracted genes in all domesticated cultivars. Interestingly, we found significant enrichment for the flavone and flavonol biosynthesis, and the isoflavonoid biosynthesis pathways in FJ, RA and JG, which produces low levels of antioxidants, which may have an association with the extent of flesh browning⁵⁵⁻⁵⁷ (Table S12). We also supplemented the information relating to the clusters in Supplemental Table S12.

7. Reviewer’s comment: *Line 176 the sentence while correct seems irrelevant. This paragraph appears to claim some positive selection (Ka/Ks) of stress response genes but the data is not presented. Table S13 appears to simply list the number of positively selected gene families, but there is no detail and no comparison of specific gene families shown, neither the legends nor the methods explain what was compared.*

Author’s response: We acknowledge the reviewer for the suggestion. We have now provided the positively selected genes detail information in Supplemental Table S13-22 and revised the text accordingly.

8. Reviewer’s comment: *The authors identify 20220 genes with CNVs. Table S14 appears to show many (most?) of these are the presence of a gene in one accession and absence in the other 12 accessions or perhaps presence in one other accession. I’m not sure this is what is normally understood as a CNV although there does not appear to be any strict definition in the*

field. I think most readers would find the observation of nearly half the genes in 13 closely related accessions defined as subject to CNV as surprising. A clear definition of what the authors are calling a CNV would be helpful to put this number into context especially since the majority of CNVs identified appeared to have no effect on expression in leaves or fruit. Are there any features of CNVs associated with expression changes that are unique and might provide some predictive value as to whether a given CNV was worth investigating further?

Author's response: Thank you to the reviewer for this suggestion. To help readers better understand the significance and number of CNVs, we added a definition in the MS Results section: 'CNVs are polymorphisms within species in which sections of a genome differ in copy number between individuals, and include deletions, duplications, or amplifications (the same sequence of DNA is duplicated multiple times, typically in tandem) of DNA sequence⁴².'

In our Results section, we identified 20,220 protein-coding CNV genes using CNVnator, which is similar to results obtained for CNV identification in rice (25,549 gCNVs). We also found that the copy number of most CNV genes, which were positively correlated with their expression levels, was less than 3, leading us to speculate that any gene with more than 3 copies may have acquired subfunctions (inducible expression under specific conditions) or have become non-functional (Table S25-27). We also added the related text in the Discussion section.

9. Reviewer's comments: *There is no method given for the transient expression studies shown in Figure 3F nor any description of the vectors used*

Author's response: We have now added the method for transient expression in the M & M section: 'Bagged 'Gala' apple fruit were harvested 90 days after bloom for transient expression studies. TRV vectors and *Agrobacterium* were constructed as previously described¹²⁸. The infiltration protocol and culturing methods for transient expression assays were adapted from previously described methods¹²⁹⁻¹³¹. The infected apples ('Gala') were placed at 23 °C for either 2 days (for overexpression) or 5 days (for silencing). Infiltration was repeated using three biological replicates.'

10. Reviewer's comment: *Line 215 Again Table S19 is poorly described and it is unclear to me what is meant by "PAV length was significantly longer..." Figure 4B shows inversions as*

being longest and translocations longer as well although the total length of all PAV SVs combined is longer. I could not understand what was meant by Line 217. I think the authors are referring to Figure 4C which shows most SVs are present in only one accession I would not normally expect this to be described as allele frequency.

Author's response: Thank you for bringing up this issue. We revised Table S28 to make the table clearer. We apologize for the incorrect description in the 'Results' and we deleted the sentence in L215 and the word 'allele'. In the section, we try to declare that PAV is the major genomic variation type, as we describe in L214.

11. Reviewer's comment: *Line 220 The statement that PAVs were concentrated on two particular chromosomes is not supported by figure 4E which shows the distribution to be relatively even. I accept that there is an uneven distribution of SVs along the chromosomes (hot spots) but not between the chromosomes.*

Author's response: Thanks for pointing this out. We have now revised the sentence in line 222 to: 'There was an uneven distribution of SVs along the chromosomes, with 123 SV hotspot regions (Table S29; Figure S8), indicating that multiple, independent SVs have arisen in these regions.'

12. Reviewer's comment: *The paragraph starting at line 222 appears to conclude that because some known genes are found in regions with SVs that they undergoing selection without any supporting data for this conclusion.*

Author's response: We have now revised the related text 'Previously SVs related to phenylpropanoid and brassinosteroid biosynthesis were reported that involved in plant development, fruit coloration, response to environment. These findings are consistent with a previously reported trend that variants harbored within SV hotspots may undergo stronger environmental selection than those in other genome regions.'

13. Reviewer's comment: *The paragraph starting a line 235 suggests a relationship between SVs and genes that results in altered expression. However, there is no definition of "altered expression". Figure S7 relates SVs in different regions to altered expression but it is not*

clear whether it is the position of the SV or the expression that is significant nor what measure of significance was used. This section needs to be clarified and expression data and statistical tests included.

Author's response: We added an explanation in the M & M section: 'SVs in the different regions were differentially expressed ($|\text{Fold Change}| \geq 2$ and $\text{FDR} < 0.01$), DESeq2 was used, in which the difference test distribution model is a negative binomial distribution¹¹⁷.'

14. Reviewer's comment: *The paragraph starting at line 265 talks about "shared SVs" it is not clear what is meant by this term. It is not clear why these nine genes have been selected from the 46k genes in the genome or how the SV is connected.*

Author's response: We revised the text to: 'According to the physiological characteristics of fruit, we divided the 13 accessions into different groups, such as disease-resistant and non-disease-resistant; colored and uncolored. We then found related SVs to these characteristics in each compared group. This revealed nine SVs present in both the disease-resistant related group and the color-related group. The 9 shared SVs associated with fruit color and disease resistance, which are potentially involved in trade-offs between fruit quality and immunity, are listed in Table S34.'

15. Reviewer's comment: *In the discussion line 277 claims the authors identified selective SVs and CNVs underlying several important traits yet this data is not present in the manuscript.*

Author's response: We revised the text to: 'We identified selective SVs and CNVs underlying fruit coloration, and found that selective SVs may also involve trade-offs between fruit quality and immunity.'

Response to Reviewer 2

Reviewer's comment: *In this manuscript Wang et al. report on the de novo assembly of 10 apple genomes (wild and cultivated ones) on of which at gold standard level. They then take advantage of these assemblies to look for gene copy number variations between the different apple accessions. The authors then find a correlation between gene copy number and expression levels. Using a candidate approach the authors then show that MdSAUR72 may play a role in*

apple fruit coloration. Similarly, they could associate *MdMMK2* with fruit coloration.

Author's response: We thank the reviewer for this suggestion. We performed the virus-induced gene silencing (VIGS) of *MdMMK2* in apple fruit to demonstrate causal effect of *MMK2*. To explore the effect of sequence variation of *MdMMK2* on gene expression we performed a firefly luciferase (LUC) complementation imaging assay in *N. benthamiana* leaves using the fourth intron of *MdMMK2* with insertion or without insertion as effector, *MdMMK2* as reporter. We have allelotyped the *MdMMK2* loci in a larger population with varied fruit color phenotypes. Thus we proved that the 209 bp insertion in the fourth intron of *MdMMK2* which may acts as a noncoding RNA to regulate the *MMK2* expression as well as fruit coloration.

1. Reviewer's comment: *Please explain in more detail how exactly the different wild and cultivated apple accessions have been selected. Were there any reasons from a genetic diversity level to select those accessions?*

Author's response: We thank the reviewer for this suggestion. We have now added phenotypic and other detailed information regarding the 13 *Malus* accessions to explain the genetic diversity of these accessions. The phenotype of the 13 *Malus* accessions showed diversity in characteristics such as color, size, stress resistance (Figure 1A). The *M. sieversii* and *M. sylvestris* genomes directly contributed to cultivated apple (*M. sieversii* is the primary progenitor and *M. sylvestris* is a major secondary contributor to cultivated apples). *M. sieversii* showed admixed ancestry possibly from hybridizations with wild apples such as *M. orientalis*. The Chinese native species, *M. asiatica* 'Zisai Pearl' has soft fruit, which are often consumed as dessert apples. Based on the genetic diversity and phenotypic diversity, we believe that the pan-genome using the 13 assembled genomes will broaden our understanding of apple speciation, differentiation, and evolution.

2. Reviewer's comment: *The approach for transposable element annotation is non-standard. Why did the authors choose this approach and not some standard pipeline such as REPET or EDTA?*

Author's response: We conducted annotation of TEs as follows: Homolog evidence and alignment searches were performed using the RepBase database (<http://www.girinst.org/replib>),

and next the RepeatProteinMask (<http://www.repeatmasker.org/>). For *de novo* annotation, LTR_FINDER⁸⁴, PILER⁸⁵, RepeatScout (<http://www.repeatmasker.org/>), and Repeat-Modeler (<http://www.repeatmasker.org/RepeatModeler.html>) were used to construct *de novo* libraries, then annotation was carried out with RepeatMasker (<http://repeatmasker.org/>). We note that several other studies have conducted TE annotations using the same methods⁹⁶⁻⁹⁸.

The principle of TE annotation mentioned above is similar to the working principle of EDTA⁹⁹.

3. Reviewer's comment: *In line 134 the authors present a range of core genes. Should that not be a single number as these genes must be in common between all accessions?*

Author's response: We apologize for the incorrect description. During pan-genome construction, gene clusters shared by all 13 accessions were defined as 'core', and then the genes in each cluster and in each cultivar were further identified and defined as 'core genes'. Therefore, the number of core gene clusters is fixed for each cultivar, while the number of core genes may vary and are presented as a range. We revised the text to: 'Through gene cluster analysis by OrthoFinder⁴⁵, 14,896 clusters that were present in all 13 accessions were defined as 'core', 6,314-7,408 clusters that occurred in 11 to 12 accessions were defined as 'softcore', 8,226-12,762 clusters that occurred in 2 to 10 accessions were defined as 'accessory (dispensable)', and 766-2,807 clusters that were represented in only one accession were defined as 'specific' (Figure 1; Table S11).'

4. Reviewer's comment: *Why have the authors only focused on gene CNV and not also transposable element CNVs? These could be very interesting too, particularly since the authors often observe CNVs in promoters. I suspect that this could be the result of a TE CNV (such as a MITE).*

Author's response: We thank the reviewer for this suggestion. We attempted to identify TE CNVs in the genome using the same method as for gene CNV identification, and found, by alignment validation, that the results were imprecise (accuracy rate \leq 20%). We supposed the reason that genes are more conserved and their copy numbers can be relatively easily distinguished by setting a threshold. The polymorphism of TEs is higher than that of genes.

Therefore, we only focused on identifying gene CNVs in this study.

5. Reviewer's comment: *Concerning the insertion at MdMMK2: what is the sequence there, is it a TE?. Also, to strengthen the correlation with fruit color, it must be tested in a larger population whether this insert is present or not and if that influences fruit color (e.g., by PCR).*

Author's response: We thank the reviewer for this suggestion. The insertion at *MdMMK2* was a LTR/Gypsy TE, and we have added this information in the Results section as: 'One of these was a 209 bp insertion (LTR/Gypsy TE) in the intron of mitogen-activated protein kinase homolog MMK2 (GS07G0097300) (Figure 5A).'

To further test the correlation between *MdMMK2* and fruit color, we also conducted a PCR-based analysis in a larger population. We have now allelotyped the *MdMMK2* locus in 24 cultivars with varied fruit color phenotypes (<https://npgsweb.ars-grin.gov/gringlobal/cropdetail.aspx?type=descriptor&id=115>). We confirmed that the cultivars with uncolored fruit were heterozygous for a 209 bp insertion in the intron of *MdMMK2*, while the cultivars with red colored fruit were homozygous without the 209 bp insertion. These results indicate that the *MdMMK2* allele is associated with fruit coloration (Figure 5E; Table S33). We added this result in the revised manuscript.

6. Reviewer's comment: *In the discussion the authors state that: "We identified selective SVs and CNVs underlying fruit size, flesh firmness, acidity, stress resistance, and found that selective SVs may also involve trade-offs between fruit quality and immunity." This statement should be toned down as the results presented here are purely correlative and this statement quite speculative.*

Author's response: We revised the text to: 'We identified selective SVs and CNVs underlying fruit coloration, and found that some evidence that selective SVs may also involve trade-offs between fruit quality and immunity.'

7. Reviewer's comment: *Finally, how were the copy numbers correlated with gene expression? How were the different haplotypes separated or were all reads mapped to one copy of the corresponding gene? Did the authors observe that all additional copies were expressed or*

were some individual copies more highly expressed?

Author's response: We used the number of copies and the amount of gene expression to analyze the correlation between CNVs and expression. It is difficult to distinguish the relationship between each copy and expression.

The relationships between copy numbers and gene expression levels were calculated using a Pearson correlation test. For copy number variation (CNV), the PacBio reads of the 13 accessions were individually aligned to the GS genome, using Minimap2 (v2.24)¹⁰⁰. Based on the alignments, CNVs were called using CNVnator (v0.4.1)¹⁰¹. To analyze gene copy number variations, we aligned the cognate proteins in the CNV regions of GS to each genome assembly using BLAT. It is very difficult to determine the expression levels of each copy when multiple copies exist; however, we found that the copy number of most CNV genes, which were positively correlated with their expression levels, was less than 3, leading us to propose that any gene with more than 3 copies may have acquired subfunctions (inducible expression under specific conditions) or have become non-functional (Table S25-27). We added related text in the Discussion section.

Minor concern

1. Reviewer's comment: *Line 71 please define gCNVs.*

Author's response: Corrected.

2. Reviewer's comment: *Line 111 does not make sense to me, does the length correspond to the sum of all TE lengths added together? Please rephrase.*

Author's response: The method for calculating the total length of TEs involves adding the lengths of TEs at different locations and removing redundant portions of those lengths. To make the sentence more clearly, we revised it to: 'We determined that the sequence length of each assembled transposable element (TE) ranged from 359,707,305 to 112,373,813,146 bp, accounting for 54.06% to 56.21% of the total assembled sequence length (Table S7; Figure S3A).'

3. Reviewer's comment: *Line 122 is a repetition of what is already mentioned in line 114 and 115.*

Author's response: We deleted the repeat sentence.

4. Reviewer's comment: *Line 299 incomplete sentence.*

Author's response: Corrected.

Response to Reviewer 3

Reviewer's comment: *The paper by Wang et al describes how they built a pangenome by integrating existing Malus genomes with newly sequenced ones, obtained with long reads. Only the Granny Smith (GS) obtained the gold standard label according to LTR integrity, with the others achieving close scores. They carry out a comprehensive analysis of the pangenome which I think can be inspiring for other people working on fruit trees. There 's room for improvement though (see my comments below), and my first request would be to make the pangenomic resources produced for this paper available to everyone. That would be the most important single feature that will increase the impact of this work.*

Author's response: We thank the reviewer for your suggestions. All sequencing data generated in this study have been submitted to the National Genomics Data Center BioProject no. PRJNA872768. The genome sequences are accessible under NCBI BioProject no. PRJNA869488 and no. PRJNA927238. We also have now performed additional data analysis and performed additional experiments to demonstrate causal effect of *MMK2* in apple to improve the manuscript.

1. Reviewer's comment: *Please scan and correct spelling throughout the text, for instance at L21, L64, L225.*

Author's response: We thank the reviewer for providing this feedback. We have corrected these errors and carefully revised the manuscript, which has now been re-edited by the professional editing company, PlantScribe.

2. Reviewer's comment: *Sentences L15, L67 and L778 are apparently contradictory, please clarify how many new assemblies were produced. After reading page 4 it seems a total of 10 genomes were assembled.*

Author's response: Thank you for pointing out this issue. We assembled high-quality

genomes of 10 diverse apple accessions, and three genome sequences (MSM, MSR, GA) were obtained from a published assembly (PRJNA591623). Thus, 13 apple genomes were used to construct the apple pan-genome. We revised the sentence in line 67 to: ‘Here, we assembled high-quality genomes of 10 diverse apple accessions that collectively exhibit broad diversity in fruit quality and disease resistance.’.

3. Reviewer’s comment: *The BUSCO scores on L95 and L104 are also in contradiction. How can core genes achieve higher BUSCO completeness (98.8 > 98.4)?*

Author’s response: We apologize for the unclear description in this part. We present the evaluation results from BUSCO in L95 and L104. To make the sentences more accurate, the text in L95 has been modified to: ‘We further evaluated the completeness of the assemblies using Benchmarking Universal Single-Copy Orthologs (BUSCO)⁴³, and found scores between 97.1% (GS) and 98.4% (FJ), which are higher than the corresponding values of a previously published *Malus* genome (HFTH1, PRJNA482033), indicating high completeness within genic regions (Table S3-S5).’; and the text in L104 to: ‘Our set of 10 apple genomes was evaluated by BUSCO, with each accession achieving a score > 95%. The scores ranged from 96.3% (MO) to 98.8% (OR) (Table S5; Figure S2A)’. Table S5 was also revised to: ‘Completeness assessment of 10 apple genomes using BUSCO (Benchmarking Universal Single-Copy Orthologs).’.

4. Reviewer’s comment: *I guess you refer to non-redundant TE sequences in L118, but please clarify.*

Author’s response: Thank you for pointing out this problem. In fact, the statistical data for non-redundant sequences was generated during the construction of the pan-genomes. We apologize for the writing error here. To avoid misunderstanding of this sentence, we have deleted it.

5. Reviewer’s comment: *L135 why core estimates are ranges instead of a single number? Please explain.*

Author’s response: We apologize for the incorrect description. During pan-genome construction, gene clusters shared by all 13 accessions are defined as ‘core’, and then the genes

in each cluster and in each cultivar were further identified and defined as ‘core genes’. Therefore, the number of core gene clusters is fixed for each cultivar, while the number of core genes may vary and are presented as a range. We revised the text to: ‘Through gene cluster analysis by OrthoFinder⁴⁵, 14,896 clusters that were present in all 13 accessions were defined as ‘core’, 6,314-7,408 clusters that occurred in 11 to 12 accessions were defined as ‘softcore’, 8,226-12,762 clusters that occurred in 2 to 10 accessions were defined as ‘accessory (dispensable)’, and 766-2,807 clusters that were represented in only one accession were defined as ‘specific’ (Figure 1; Table S11).’.

5.1 Reviewer’s comment: *Figure 1A: please justify why families are used instead the usual gene clusters (as those in L134). In my opinion a family-based analysis assumes that all each family corresponds roughly to a biological function, which is clearly an over-simplification. In fact see L293.*

Author’s response: Thank you for the helpful suggestion. We changed ‘families’ to ‘clusters’ and made more detailed divisions within the clusters. We have rewritten the text in L134 to: ‘Through gene cluster analysis by OrthoFinder⁴⁵, 14,896 clusters that were present in all 13 accessions were defined as ‘core’, 6,314-7,408 clusters that occurred in 11 to 12 accessions were defined as ‘softcore’, 8,226-12,762 clusters that occurred in 2 to 10 accessions were defined as ‘accessory (dispensable)’, and 766-2,807 clusters that were represented in only one accession were defined as ‘specific’ (Figure 1; Table S11).’

5.2 Reviewer’s comment: *Figure 1C: was the genome order shuffled to produce this figure? That's how Tettelin et al did it to avoid implicit biases.*

Author’s response: We thank the reviewer for this suggestion. We added the ‘number of genomes’ below the figure to avoid implicit biases, as in Tettelin et al.

5.3 Reviewer’s comment: *Figure 1C: a Tettelin-like function would allow you to estimate the core size in the limit; indeed the data with 13 genomes suggests the core will shrink a bit more if more genomes are added (compare this to L137).*

Author’s response: Tettelin’s method involves predicting the number of core genes by

fitting a curve. Our study results indicate that when the number of genomes was 13, the core genes had almost reached a plateau stage. We predict that increasing the number of apple genomes may not significantly increase the number of core genes. However, the pan-genome has not yet reached a plateau, suggesting high diversity among apple accessions.

5.4 Reviewer's comment: *Please make the pangenome clusters available for download so that readers can use them straight away.*

Author's response: Thank you for your suggestion. The related cluster data have now been uploaded to figshare and have the download link is: <https://figshare.com/s/4e1ea61459393ff54684>.

6. Reviewer's comment: *In the context of L131-132 please check the papers by Michele Morgante where term dispensable is discussed. I know this is only terminology but I advocate for the term accessory instead. Other people use the 'shell' term.*

Author's response: Thanks for this suggestion. A series papers by Michele Morgante inspired us to have a deep understanding of 'dispensable clusters' which dispensable genome play important in determining plant phenotypes and shaping genome evolution. Also from our cross-analysis of PAV and OrthoFinder gene clusters which confirmed that non-core genes may play key role in plant phenotypes and shaping genome evolution. We agree that the dispensable genome is less dispensable than previously thought. Considering that most papers use the terminology 'dispensable', we have now use 'accessory (dispensable)' instead of the 'dispensable'.

7. Reviewer's comment: *I believe the section from L151 needs refocus. What I mean is that the phylogeny all the way up to Arabidopsis is probably not that relevant for this paper. Instead, the new data presented here would allow to make a high-resolution phylogeny of the Malus genus and the cultivars. Of course that would require trees with bootstrap or equivalent branch support scores, which are not currently shown.*

Author's response: Thank you for this suggestion. Based on your advice, we have removed species such as *Arabidopsis thaliana* and kept only the 13 species from our pan-genome

and four other *Malus* genus species (*Pyrus sorotina*, *Prunus armeniaca*, *Prunus persica*, and *Fragaria vesca*). We have reconstructed the evolutionary tree and added bootstrap values, and the result is shown in Figure 2.

8. Reviewer's comment: *I feel the same about the CAFE analysis of family expansions. I believe is of little interest as currently shown.*

Author's response: Thank you for highlighting a potential area of concern. We conducted the enrichment analysis of expanded and contracted genes in all cultivated apples and added related text in the MS: 'To improve the understanding of the significance of expanded and contracted gene clusters, we conducted KEGG analysis of expanded and contracted genes in each domesticated cultivar. Interestingly, we found significant enrichment of the flavone and flavonol biosynthesis, and the isoflavonoid biosynthesis pathways in FJ, RA and JG (Table S12). We propose that these two pathways may confer upon these three cultivars with low levels of antioxidant activity^{132, 133}, which may have an association with the extent of flesh browning⁵⁵⁻⁵⁷.'. We also supplemented the cluster information in Supplemental Table S12.

9. Reviewer's comment: *Please explain what you mean with unbiased alignment on L188.*

Author's response: Thanks for pointing out this problem. We deleted the word 'unbiased'.

10. Reviewer's comment: *Please explain how similar two genes have to be considered 'CNV genes'. I guess on Figure 3 the expression values of all copies are added up? Are the same primers used for the qRT-PCR experiments?*

Author's response: To help readers better understand the significance and number of CNVs, we added a definition in the MS Results section: 'CNVs are polymorphisms within species in which sections of a genome differ in copy number between individuals, and include deletions, duplications, or amplifications (the same sequence of DNA is duplicated multiple times, typically in tandem) of DNA sequence⁴².'

Actually, we used the number of copies and the amount of gene expression to analyze the correlation between CNV and expression. It is difficult to split the relationship between each copy and expression. We used the same primers for RT-qPCR experiments.

11. Reviewer's comment: *Have the authors cross-checked whether the PAVs overlapping genes are confirmed in the OrthoFinder gene clusters? That would allow them to do a internal quality control. This would work both ways: gene clusters might lend support to called PAVs, and the otehr way around. This would also raise the standard of this study and will make the most of the assemblies at hand*

Author's response: Thanks for your helpful suggestions, we have now performed a cross-analysis of PAV and OrthoFinder gene clusters. As shown in Figure 4, PAVs is the largest variation type within the SVs (~80%), so we examined the content of SVs in different gene clusters. The results showed that there were few core genes containing PAVs/SVs in the exonic regions, while the non-core genes contained a high proportion of PAVs. The cross-analysis confirmed that PAVs could affect the production of non-core gene clusters, which suggest non-core genes may play key role in plant phenotypes and shaping genome evolution.

12. Reviewer's comment: *On L228 'enriched' probably means 'significantly accumulate', am I right?*

Author's response: We revised 'enriched' to 'significantly accumulate'.

13. Reviewer's comment: *Can you please compute the numbers of PAVs in promoters with different distance thresholds? There are published reports that suggest that promoters in other fruit crops (peach, for instance) are mostly conserved up to 500bp. Of course there are documented cases of conserved regulatory sequences several Kb away, but the genome-wide trend is that proximal promoters are much shorter than 5kb. In fact, the RNAseq expression data would allow to estimate this with some precision, by looking only at genes with altered expression patterns (L246).*

Author's response: Thank you for your suggestion. As suggested, we calculated the distribution of SVs upstream and downstream from genes, and obtained similar results as in peach. We added the related sentence in L246; 'The distribution of SVs upstream and downstream from genes indicated that regions closer to genes are more conserved, while regions farther away from genes more have increased frequencies of mutations' (Figure S10).

14. Reviewer's comment: *Please make the PAVs available to readers, this will accelerate apple research and will increase the profile of this paper.*

Author's response: Thank you for your suggestion. The related PAV data were uploaded to figshare and the download link is <https://figshare.com/s/c6734dc864c07f51df9c>.

15. Reviewer's comment: *Please rephrase L256, as that kinase has been linked to salt and copper stress in *A. thaliana*, not in *Malus*, where it could have different functions. In fact, the whole section about MMK2 is quite interesting but unfortunately no mechanistic model is provided to link the expression of this protein and anthocyanin content. Figure 5 shows that MMK2 could be an excellent marker for anthocyanin content nevertheless.*

Author's response: Thank you for pointing out this issue. We have rewritten the text in L256 as: 'homologs of MMK2 in *Arabidopsis thaliana* have previously been demonstrated to be upregulated in response to salt and copper stress^{70, 71}.

To test the correlation between *MdMMK2* and fruit color, we conducted PCR analysis of a larger population. to test whether this locus contributes to fruit color We have now allelotyped the *MdMMK2* loci of 24 cultivars with varied fruit color phenotypes (<https://npgsweb.ars-grin.gov/gringlobal/cropdetail.aspx?type=descriptor&id=115>). We confirmed that the cultivars with uncolored fruit were heterozygous for a 209 bp insertion in the intron of *MdMMK2*, while the cultivars with red colored fruit were homozygous without the 209 bp insertion (Figure 5E). These results indicated that the *MdMMK2* allele is associated with fruit coloration. We have now performed the virus-induced gene silencing (VIGS) of *MdMMK2* in apple fruit, further suggesting that *MMK2* functions in apple coloration and *MMK2*-silenced apple fruits exhibited reduced anthocyanin contents relative to the empty vector controls (Fig. S11F-G). A previous study showed that the second intron of *AG* encodes several ncRNAs that repress *AG* expression, so we speculate that the 209 bp insertion in the fourth intron of *MdMMK2* may acts as a ncRNA. To determine intron mRNA levels, we cloned the intron sequence from apple peel cDNA. Interestingly, the fourth intron of *MdMMK2* was detected as ncRNA. To avoid genomic DNA contamination, RNAs without RT were interrogated by PCR as a control. To explore the effect of sequence variation of *MdMMK2* on gene expression we performed a firefly luciferase (LUC)

complementation imaging assay in *N. benthamiana* leaves using the fourth intron of *MdMMK2* with insertion (*MMK2-In4⁺*) or without insertion (*MMK2-In4*) as effector, *MdMMK2* as reporter. Co-expression of the fourth intron with TE insertion of *MdMMK2* and *MdMMK2* represses *MdMMK2* transcription comparing with the co-expression of the fourth intron without TE insertion of *MdMMK2* and *MdMMK2* (Fig. 5G-J). Here, we propose the model that the intronic ncRNAs from *MdMMK2* intron 4 confer *MdMMK2* expression and may participate in anthocyanin accumulation in apple fruit (Fig. 5F). We have added this result in the revised manuscript.

16. Reviewer's comment: *L351 The description of the method of genome size estimation is clearly insufficient, more details are needed if reader are to be able to reproduce this.*

Author's response: Thanks for the helpful suggestion, we supplemented the manuscript with the detailed method for genome size estimation: 'Flow cytometry was first conducted to estimate genome size. FJ and GS leaves were collected and analyzed using a CyFlow Space Flow Cytometer (Sysmex Europe GmbH, Norderstedt, Germany), equipped with a UV-LED source (with emission at 365 nm) and a blue solid-state laser ($k = 455$ nm). GA ($2n = 2x = 24$) was used as the reference. The genome size was estimated by *k*-mer frequency analysis. The distribution of *k*-mers depends on the characteristic of the genome and follows a Poisson's distribution. Before assembly, the 17-mer distribution of CCS reads was generated using Jellyfish (v2.2.6)¹²⁵'. This was added to the M & M section.

17. Reviewer's comment: *L365 Why was embryophita used instead of eudicots for BUSCO analysis? Surely the latter makes more sense for Malus.*

Author's response: Thank you for the helpful suggestion. As the embryophyta database includes data from eudicots, so we used embryophita for BUSCO analysis.

18. Reviewer's comment: *FigureS2B: please indicate how the plot was produced.*

Author's response: Thank you for your suggestion. We have added the legend to Figure S2B: 'Genomic synteny between homologous chromosomes (haplomes) of the diploid assemblies were constructed using *MCSanX*. Numbers indicate chromosomes.'. We also revised the text in the M & M section to: 'Genomic synteny for each alignment was used to build whole-genome

synteny between GS and the other nine *Malus* genomes using *MCSanX*⁹³.

19. Reviewer's comment: *L458 Please indicate where readers can obtain the GFF files and how isoforms of the same gene were treated.*

Author's response: Thank you for your suggestion. The related GFF files were uploaded to figshare and the download link is: <https://figshare.com/s/5fea32dd0fba11ead6bd>.

20. Reviewer's comment: *Please correct reference 43 in L98*

Author's response: We revised the reference in L198.

Reviewers' Comments:

Reviewer #1:

Remarks to the Author:

Overall I found the manuscript much improved and clearer. I believe the data presented will be of significant interest to the community.

I have only one issue that I believe needs to be corrected

Line 123 clarifies that samples of fruit peel and fruit flesh were only taken from five accessions (GS, OR, SD, FJ and GA). However, it is not made clear later in the manuscript (line 195 and figure 3B) that the relationship between CNVs and fruit gene expression was examined only for those five accessions. This needs to be specified in the text and figure. It is also not clear whether leaf expression was correlated with CNVs for only those five accessions or with all the accessions.

The number of total CNVs shown for leaf and fruit samples in 3B is also the same, I am concerned that this is the total number of CNVs for ALL accessions which is an incorrect comparison. Specifically, the total number of CNVs for the fruit flesh and fruit peel comparisons in Figure 3B should be different to the total number of CNVs for leaves and should reflect the total CNVs for only those five accessions where expression was analysed in fruit.

Minor points

Figure S9 Specify which test was used to determine "Significant"

Figure S10 The Y axis is unclear. "Occurrence (x 100%)" is confusing. I'm not exactly sure what it means - I think it means the percentage of gene spaces measured where an SV occurs. If so, we need the window used and the total number of genes examined from which the percentage is calculated.

Figure 2 and associated text. Just a comment that bootstrap values below 60% are really not informative and are usually ignored

Line 282 AG should be expanded to Agamous in the first usage

The paragraph describing the experiment in Figure 5 would be clearer if the text referred to the figure more frequently. In particular line 285 should refer to 5F. And the figure legend does not describe 5F clearly at all and needs to be clarified.

Paragraph starting at line 129

All three reviewers were confused by the range of values for clusters. I think I understand that the range refers to the number of genes in the soft core class which varies between accessions while the number of soft core clusters should be a fixed number. However, the manuscript does not make that clear - to me at least.

For consistency in figure 1D, Families should be changed to clusters.

Reviewer #2:

Remarks to the Author:

In this revised version of the manuscript the authors nicely improved it and have addressed my main comments. I only have a few minor comments:

Line 63: This sentence sounds too superfluous. Of course other species will have different genomes, please clarify which aspects of a genome differs between self-compatible and self-incompatible plants.

Line 114: I think the authors mean the sum of the sequence length of assembled TE classes? Like this the sentence is wrong, no TE is that large.

Line 122: 56.21% of the genome (Table 1)

Line 125: 6 Gbp

Line 136: until a total of ten pan-genomes was included.

Line 151: I guess the authors mean stress response?

Paragraph at Line 224: Is there a specific TE family that is the main contributor to PAVs? This could be highly interesting and establish that TE as the main driver in apple evolution.

Line 272: Two times "suggesting" in the same sentence

Fig. 4E: This graph would be more telling if the number of PAVs was adjusted by the chromosome size. Maybe by indicating number of PAVs per Mb?

Reviewer #3:

Remarks to the Author:

The authors have responded to all of my previous suggestions/comments and I am satisfied with most of them.

I still have a couple of comments though, I mark with >> my previous comments and with > the responses:

>>5.1 Reviewer's comment: Figure 1A: please justify why families are used instead the usual gene clusters (as those in L134). In my opinion a family-based analysis assumes that all each family corresponds roughly to a biological function, which is clearly an over-simplification. In fact see L293.

>Author's response: Thank you for the helpful suggestion. We hanged 'families' to 'clusters' and made more detailed divisions within the clusters. We have rewritten the text in L134 to: 'Through gene cluster analysis by OrthoFinder, 14,896 clusters that were present in all 13 accessions were defined as 'core' [...]

What I meant is that OrthoFinder clusters, computed on the grounds protein similarity, are probably producing gene families in some cases, with paralogues included, instead of gene clusters. That's why other pangenome building approaches use nucleotide sequences instead. Anyway, if my suspicion is correct you should see many core clusters containing several genes from the same species, and that would explain why the core compartment is relatively small (~15K). This should be easy to check, please do it and discuss it in the text if needed.

>> 17. Reviewer's comment: L365 Why was embryophita used instead of eudicots for BUSCO analysis? Surely the latter makes more sense for Malus.

> Author's response: Thank you for the helpful suggestion. As the embryophyta database includes data from eudicots, so we used embryophita for BUSCO analysis.

In my experience using too broad BUSCO lineages (embryophita vs eudico) might overestimate completeness. In fact, looking at https://busco.ezlab.org/list_of_lineages.html you can see:

```
Lineage Nb of BUSCO markers
embryophyta_odb10.2019-11-20 1614
eudicots_odb10.2019-11-20 2326
...
fabales_odb10.2019-11-20 5366
```

It is for this reason that I would ask the authors to repeat the BUSCO estimates at the eudicot level, which is more specific for the Rosales, to which Malus belongs.

Manuscript Number: NCOMMS-23-02951B

MS TITLE: Pan-genome analysis of 13 diverse *Malus* accessions reveals structural and sequence variations associated with fruit traits.

Dear Reviewers,

We thank you for reviewing our manuscript, entitled “**Pan-genome analysis of 13 diverse *Malus* accessions reveals structural and sequence variations associated with fruit traits**”, and for your valuable suggestions. Detailed point-by-point responses to the issues raised by each reviewer are included below. Revisions are shown in red in the revised version of the manuscript. We thank you again for your time in providing thoughtful feedback and we hope the revision meets with your approval.

Yours sincerely,

Ting Wu

College of Horticulture

China Agricultural University

Reviewer 1

1. Line 123 clarifies that samples of fruit peel and fruit flesh were only taken from five accessions (GS, OR, SD, FJ and GA). However, it is not made clear later in the manuscript (line 195 and figure 3B) that the relationship between CNVs and fruit gene expression was examined only for those five accessions. This needs to be specified in the text and figure. It is also not clear whether leaf expression was correlated with CNVs for only those five accessions or with all the accessions. The number of total CNVs shown for leaf and fruit samples in 3B is also the same, I am concerned that this is the total number of CNVs for ALL accessions which is an incorrect comparison. Specifically, the total number of CNVs for the fruit flesh and fruit peel comparisons in Figure 3B should be different to the total number of CNVs for leaves and should reflect the total CNVs for only those five accessions where expression was analysed in fruit.

Response: We thank the reviewer for pointing out this issue. We have added detailed information regarding the relationship between CNVs and fruit gene expression in the main text and in the indicated figure legend. The main text now reads: “In addition, 418, 405 and 775 CNV genes were positively correlated with their expression levels in the fruit peel, fruit flesh (for accessions GS, OR, SD, FJ and GA) and leaves (for all 13 accessions), respectively (Fig. 3B; Table S25-27).” We have corrected the number of CNVs shown for the fruit, retaining only the 8779 CNVs present in accessions GS, OR, SD, FJ, and GA (Table S23) and redrawn Fig. 3B to reflect this.

Minor points:

1. Figure S9 Specify which test was used to determine “Significant”

Response: Significance was determined with Student’s *t*-test at $p < 0.05$. This information has been added to the legend for Figure S9.

2. Figure S10 The Y axis is unclear. “Occurrence (x 100%)” is confusing. I’m not exactly sure what it means – I think it means the percentage of gene spaces measured where an SV occurs. If so, we need the window used and the total number of genes examined from which the percentage

is calculated.

Response: We have revised the Y axis label to read “Percentage of sequences containing SVs (%)”. There were 46,050 genes in this analysis and we analyzed the distribution of SV breakpoints in 10 bp windows. This information has been added to the main text (line 262).

3. *Figure 2 and associated text. Just a comment that bootstrap values below 60% are really not informative and are usually ignored.*

Response: We thank the reviewer for raising this point. We have deleted bootstrap values below 60% in Figure 2B.

4. *Line 282 AG should be expanded to Agamous in the first usage*

Response: We have used the full gene name (*AGAMOUS*) in the first instance prior to introducing the gene symbol in the revised manuscript.

5. *The paragraph describing the experiment in Figure 5 would be clearer if the text referred to the figure more frequently. In particular line line 285 should refer to 5F. And the figure legend does not describe 5F clearly at all and needs to be clarified.*

Response: Thank you for this suggestion. We have included additional references to the figure in the main text (lines 290 and 292) and edited the legend for Figure 5F as suggested to include the following: “Total RNA (i.e., RNA that was not reverse transcribed) was used as the template in standard PCR reactions as a control. The lack of a band indicated that no genomic DNA remained. The last four lanes show reactions in which apple peel cDNA was used as the template and the primers were specific for the *MdMMK2* fourth intron.”

6. *Paragraph starting at line 129All three reviewers were confused by the range of values for clusters. I think I understand that the range refers to the number of genes in the soft core class*

which varies between accessions while the number of soft core clusters should be a fixed number. However, the manuscript does not make that clear – to me at least.

Response: We apologize for the earlier lack of clarity regarding the definitions used. We here classified clusters present in 11 or 12 accessions as softcore clusters; 7,786 clusters (16.78% of the total clusters) were classified as softcore clusters, which contained 123,759 genes (20.95% of the total gene number). The range refers to the total number of clusters in each of the 13 accessions (6,314–7,408). We have revised the text to clarify this point.

7. For consistency in figure 1D, Families should be changed to clusters.

Response: This has been changed as suggested.

Reviewer 2

1. *Line 63: This sentence sounds too superfluous. Of course other species will have different genomes, please clarify which aspects of a genome differs between self-compatible and self-incompatible plant.*

Response: We thank the reviewer for pointing this out and have revised the sentence consistent with your suggestion (lines 55–56).

2. *Line 114: I think the authors mean the sum of the sequence length of assembled TE classes? Like this the sentence is wrong, no TE is that large.*

Response: We thank the reviewer for pointing out this problem. Indeed, we meant the sum of the sequence length of the assembled TEs. The relevant section has been corrected in the main text.

3. *Line 122: 56.21% of the genome (Table 1)*

Response: This has been revised as suggested.

4. *Line 125: 6 Gbp*

Response: This has been corrected in the revised text.

5. Line 136: until a total of ten pan-genomes was included.

Response: This has been edited as suggested.

6. Line 151: I guess the authors mean stress response?

Response: We thank the reviewer for pointing this out. We have revised the indicated sentence, which now reads: “These findings highlight a remarkable difference in composition of secondary metabolites between wild and cultivated apples which may explained the varied stress response between wild and cultivated apples.”.

7. Paragraph at Line 224: Is there a specific TE family that is the main contributor to PAVs? This could be highly interesting and establish that TE as the main driver in apple evolution.

Response: We thank the reviewer for this suggestion. We determined that 71.31% of PAVs formed by TEs, which has higher LTR/Gypsy TE proportion (43.27%) (Fig. S8). This result was consistent with previous studies showing that evolution of LTR-RTs have created abundant genetic diversity among species⁶⁴.

8. Line 272: Two times "suggesting" in the same sentence

Response: The text has been revised as follows to avoid this redundancy: “A transgenic assay also suggested that major *MMK2* expression promoted anthocyanin accumulation in apple calli, further indicating that *MMK2* functions in apple coloration and *MMK2*-silenced apple fruits exhibited reduced anthocyanin contents relative to the empty vector controls.”

9. Fig. 4E: This graph would be more telling if the number of PAVs was adjusted by the chromosome size. Maybe by indicating number of PAVs per Mb?

Response: We thank the reviewer for this helpful suggestion. We have altered Fig. 4E to show the PAV distribution across a map of the apple reference genome, demonstrating the number of PAVs per Mb.

Reviewer 3

1. I still have a couple of comments though, I mark with >> my previous comments and with > the responses:

>>5.1 Reviewer's comment: Figure 1A: please justify why families are used instead the usual gene clusters (as those in L134). In my opinion a family-based analysis assumes that all each family corresponds roughly to a biological function, which is clearly an over-simplification. In fact see L293.

>Author's response: Thank you for the helpful suggestion. We hanged 'families' to 'clusters' and made more detailed divisions within the clusters. We have rewritten the text in L134 to: 'Through gene cluster analysis by OrthoFinder, 14,896 clusters that were present in all 13 accessions were defined as 'core'[...]

What I meant is that OrthoFinder clusters, computed on the grounds protein similarity, are probably producing gene families in some cases, with paralogues included, instead of gene clusters. That's why other pangenome building approaches use nucleotide sequences instead. Anyway, if my suspicion is correct you should see many core clusters containing several genes from the same species, and that would explain why the core compartment is relatively small (~15K). This should be easy to check, please do it and discuss it in the text if needed.

Response: We thank the reviewer for raising this point and have determined the number of genes in each core cluster as suggested. We classified clusters shared by all 13 accessions as core clusters, of which there were 14,896 (32.10% of the total number of clusters) containing 287,868 genes (48.72% of all genes). The fact that ~1/3 of all clusters were core clusters but closer to half of all genes in each genome were present in core clusters indicated the presence of multi-copy genes derived from whole genome duplication. This information has been added to the revised manuscript. Furthermore, we constructed a pan-genome using nucleotide sequence-based methods to confirm the number of core genes. Using this method, we identified 22,005 gene loci that were retained in all 13 genomes. These still comprised approximately half of each genome and were termed the core loci.

2. >> 17. Reviewer's comment: L365 Why was embryophita used instead of eudicots for

BUSCO analysis? Surely the latter makes more sense for Malus.

> Author's response: Thank you for the helpful suggestion. As the embryophyta database includes data from eudicots, so we used embryophita for BUSCO analysis.

In my experience using too broad BUSCO lineages (embryophita vs eudico) might overestimate completeness. In fact, looking at https://busco.ezlab.org/list_of_lineages.html you can see:

Lineage Nb of BUSCO markers

embryophyta_odb10.2019-11-20 1614

eudicots_odb10.2019-11-20 2326

...

fabales_odb10.2019-11-20 5366

It is for this reason that I would ask the authors to repeat the BUSCO estimates at the eudicot level, which is more specific for the Rosales, to which Malus belongs.

Response: We thank the reviewer for this feedback. We have now used the eudicots_odb10 database for BUSCO analysis, and found scores between 96.9% (GS) and 98.0% (FJ), (97.1% (GS) and 98.4% (FJ) at embryophita level), each accession achieving a score > 95%. The scores ranged from 95.4% (MO) to 98.5% (OR) (96.3% (MO) to 98.8% (OR) at embryophita level). We also used the results of eudicots level for BUSCO analysis instead of the embryophita level in the revised manuscript. (lines 91, 100 and 408; Fig. S2A; Table 1, Table S5).

Reviewers' Comments:

Reviewer #1:

Remarks to the Author:

I have no further comments and I am happy to see the manuscript published

Reviewer #2:

Remarks to the Author:

The authors have fulfilled all my requests. I have checked the manuscript and have no additional comments.

Reviewer #3:

Remarks to the Author:

I thank the authors for their efforts and improvements on the paper. My only remaining suggestion is that they mention their nt core genome size estimate, ie 22,005, so that readers can put the numbers in this section in context.

Manuscript Number: NCOMMS-23-02951C

MS TITLE: Pan-genome analysis of 13 diverse *Malus* accessions reveals structural and sequence variations associated with fruit traits.

Dear Reviewers,

We thank you for reviewing our manuscript, entitled “**Pan-genome analysis of 13 diverse *Malus* accessions reveals structural and sequence variations associated with fruit traits**”, and for your valuable suggestions. Detailed point-by-point responses to the issues raised by each reviewer are included below. Revisions are shown in red in the revised version of the manuscript. We thank you again for your time in providing thoughtful feedback and we hope the revision meets with your approval.

Yours sincerely,

Ting Wu

College of Horticulture

China Agricultural University

Reviewer 3

1. *I thank the authors for their efforts and improvements on the paper. My only remaining suggestion is that they mention their nt core genome size estimate, ie 22,005, so that readers can put the numbers in this section in context.*

Response: Thank you for this suggestion. We have added the number of core genes loci (22,005) using nucleotide sequence-based methods in the main text (Line 141-144).